# Transition metal-like carbocatalyst

Zhicheng Luo [1,4], Renfeng Nie[2,4], Vy T. Nguyen [3], Abhranil Biswas[2], Ranjan K. Behera[2], Xun Wu[2], Takeshi Kobayashi[1], Aaron Sadow [1,2], Bin Wang [3✉], Wenyu Huang [2✉] & Long Qi [1✉]

Catalytic cleavage of strong bonds including hydrogen-hydrogen, carbon-oxygen, and carbon-hydrogen bonds is a highly desired yet challenging fundamental transformation for the production of chemicals and fuels. Transition metal-containing catalysts are employed, although accompanied with poor selectivity in hydrotreatment. Here we report metal-free nitrogen-assembly carbons (NACs) with closely-placed graphitic nitrogen as active sites, achieving dihydrogen dissociation and subsequent transformation of oxygenates. NACs exhibit high selectivity towards alkylarenes for hydrogenolysis of aryl ethers as model bio-oxygenates without over-hydrogeneration of arenes. Activities originate from cooperating graphitic nitrogen dopants induced by the diamine precursors, as demonstrated in mechanistic and computational studies. We further show that the NAC catalyst is versatile for dehydrogenation of ethylbenzene and tetrahydroquinoline as well as for hydrogenation of common unsaturated functionalities, including ketone, alkene, alkyne, and nitro groups. The discovery of nitrogen assembly as active sites can open up broad opportunities for rational design of new metal-free catalysts for challenging chemical reactions.

[1] U.S. DOE Ames Laboratory, Iowa State University, Ames, IA 50011, USA. [2] Department of Chemistry, Iowa State University, Ames, IA 50011, USA. [3] School of Chemical, Biological and Materials Engineering, University of Oklahoma, Norman, OK 73019, USA. [4] These authors contributed equally: Zhicheng Luo, Renfeng Nie. ✉email: wang_cbme@ou.edu; whuang@iastate.edu; lqi@iastate.edu

Hydrodeoxygenation of oxygenates with molecular hydrogen is critical for upgrading emerging feedstocks like biomass[1,2]. Efficacious hydrodeoxygenation heavily relies on heterogeneous catalysts based on transition metals for hydrogenolysis and hydrogenation[3], although over-hydrogenation of functionalities like arenes is often encountered[4]. More sustainable and selective metal-free catalysts could overcome these limitations and are highly appealing, especially when metal contamination in products can be an issue[5], but their design and deployment remains a formidable task in both hydrogenolysis of C–O linkages and hydrogenation/dehydrogenation.

Metal-free catalysts have been reported primarily for hydrogenation[6] and oxidative dehydrogenation[7,8] but not for hydrogenolysis and non-oxidative dehydrogenation. For instance, frustrated Lewis pairs (FLPs) are capable of directly reducing olefins or imines with $H_2$[9,10]; however, the majority of FLPs are easily deactivated in the presence of water and alcohols[11,12], preventing their use in industrial applications. Most other metal-free catalysts cannot directly employ $H_2$ but instead use its surrogates like hydrazine or $NaBH_4$[13–15]. Carbon materials catalyze the hydrogenation of the π bonds of ethylene and acetylene, but not for the breaking of stronger bonds (e.g., C–O and C–H)[16]. Metal-free catalysts have shown activities in dehydrogenation but only with the assistance of oxidants.

Here, we present a carbon-based catalyst for hydrogenolysis of C–O linkages in aromatic oxygenates using molecular hydrogen. The unexpected, unique reactivity present in this study primarily originates from the nitrogen assemblies introduced by the diamine precursors into the graphitic carbons, which also enables the non-oxidative dehydrogenation of ethylbenzene and tetrahydroquinoline (THQ) and the selective hydrogenation of unsaturated functional groups in aromatics. Handling these metal-free catalysts does not require oxygen- and moisture-free conditions.

## Results and discussions

**Synthesis and characterization.** We synthesized N-assembly carbons (NACs) with closely placed nitrogen sites by condensing ethylenediamine (EDA) and carbon tetrachloride by a hard-template route[17,18] (Fig. 1a), followed by carbonization under flowing Ar at 600–900 °C and then etching to remove the silica template, giving the materials NAC-600–NAC-900. In the first step, the diamine-carbon tetrachloride condensation results in irreversible formation of C–N bonds, as evidenced by the presence of secondary amines formed as crosslinked acyclic chains and imidazolidine, shown by dynamic nuclear polarization (DNP)-enhanced $^{15}N\{^1H\}$ cross-polarization magic-angle spinning (CPMAS) NMR (Fig. 1b and Supplementary Fig. 1)[19,20]. Upon carbonization at 300 °C for 10 min, an additional broad signal appeared in the range between −180 and −230 ppm in the $^{15}N\{^1H\}$ CPMAS spectrum (Fig. 1e). Fast CP build-up suggests that the new signal can be assigned to the N–H-containing species, imidazolium[19], in which the aromatic ring contains two nitrogen atoms (Supplementary Fig. 2). At higher temperatures (>600 °C), these closely placed nitrogen atoms were further incorporated into NACs. The graphitic nature of NACs is shown in the corresponding powder X-ray diffraction (XRD) patterns and Raman spectra (Supplementary Figs. 3 and 4). The thermal stability of the NACs is evidenced by thermogravimetric analysis (TGA) (Supplementary Fig. 5) and in situ diffuse reflectance infrared Fourier transform spectroscopy (DRIFTS) study (Supplementary Fig. 6a, b).

NACs, isolated after silica etching, possess ordered mesopores (Fig. 1c and Supplementary Fig. 7)[21], uniform particle sizes (Supplementary Fig. 7), and 9.0–19.8 at% N among carbon skeleton. Element mapping (Fig. 1d) indicates that the N atoms are homogeneously distributed over the bulk of the carbon, agreeing well with the X-ray photoelectron spectra (XPS) and elemental analysis (Supplementary Table 1). High-resolution N1$s$ XPS studies (Fig. 1f, Supplementary Fig. 8 and Supplementary Tables 2, 3) resolved pyridinic, pyrrolic, graphitic, and pyridine N-oxide species in NACs[22]. Upon carbonization, the signals in N1$s$ XPS corresponding to pyridinic and pyrrolic N content decrease significantly (Supplementary Tables 2 and 3). The trend of graphitic N follows a volcanic profile, the maximum of which reaches 4.8 at% among all atoms in NAC-800 (Supplementary Fig. 9). The NAC catalysts were prepared using high-purity precursors from different sources to ensure that the carbocatalysts were entirely metal-free (Supplementary Table 4). Besides, signals for other elements, particularly common metal contaminants (e.g., Cu, Co, Ni, Ru, Pd, Pt, and Rh), were below detection limits of ICP-MS, scanning transmission electron microscope (STEM)-EDX, and XPS (Supplementary Figs. 10 and 11).

Dissociative chemisorption of $H_2$ is a prerequisite step in catalytic hydroprocessing. We demonstrated that NAC-800 dissociates $H_2$ using ambient-pressure pulsed $H_2$ chemisorption at elevated temperatures (60–360 °C), which shows a remarkable temperature dependence (Fig. 1g and Supplementary Fig. 12). A small, but reproducible chemisorption value of ~0.5-μmol $H_2$ g$^{-1}$ at 60 °C increases nearly 12-fold upon elevating the temperature to 240 °C to 5.9-μmol $H_2$ g$^{-1}$. The $H_2$ chemisorption is significantly suppressed to 1.5-μmol $H_2$ g$^{-1}$ and ultimately zero at 300 and 360 °C, respectively. In situ DRIFTS study showed the formation of C–H bonds rather than O–H or N–H at 240 °C under the flow of $H_2$ (Supplementary Fig. 6c, d).

**Catalytic studies.** The metal-free NAC materials catalyze hydrogenolysis of C–O bonds in 2-phenoxy-1-phenylethan-1-ol (PPE, **1**) with 20-bar $H_2$ (Fig. 2a), which typically require transition metal catalysts. PPE, containing both α-OH and phenolic ether, is considered a model compound for the β-O-4 linkage in lignin[23]. The NAC-800 catalyst gives the highest conversion (30%), compared with NAC-650 (7%), NAC-700 (15%), and NAC-900 (25%) in experiments performed in 2-propanol (2-PrOH) for 8 h at 230 °C, while NAC-600 is inactive (Fig. 2b and Supplementary Table 5). Quantitative conversion of PPE is achieved after 80 h, with phenol and ethylbenzene formed as the major final products (Supplementary Fig. 13). No PPE conversion was observed in the control experiments with control catalysts containing common metal contaminants (such as Fe, Cu, and Ni in Supplementary Table 6).

Three intermediates—styrene, 1-phenylethanol, and acetophenone—are detected in experiments employing the NAC-800 catalyst. The reaction network for PPE conversion over NAC-800 catalyst is shown in Fig. 2a. The yield of styrene peaks at 4 h (13%), whereas the 1-phenylethanol and acetophenone increases until 16 (7%) and 40 h (5%) of reaction time, respectively (Supplementary Fig. 13). Independent experiments show that NAC-800 catalyzes conversion of both styrene and 1-phenylethanol to ethylbenzene. The intermediate styrene is present in the reaction mixture until all PPE are consumed. Specifically, styrene and 1-phenylethanol are formed directly from PPE, whereas ethylbenzene and acetophenone are daughters. The acetophenone most likely form through equilibrium with 1-phenylethanol, suggesting that the catalyst is active for both hydrogenation and dehydrogenation in ketone–alcohol conversion. At longer reaction times, acetophenone undergoes hydrogenation to 1-phenylethanol and then hydrogenolysis to ethylbenzene. In addition, phenethoxybenzene (PEB), resulting from direct hydrogenolysis of $C_\alpha$–OH in PPE, is a terminal product that continuously increases over the

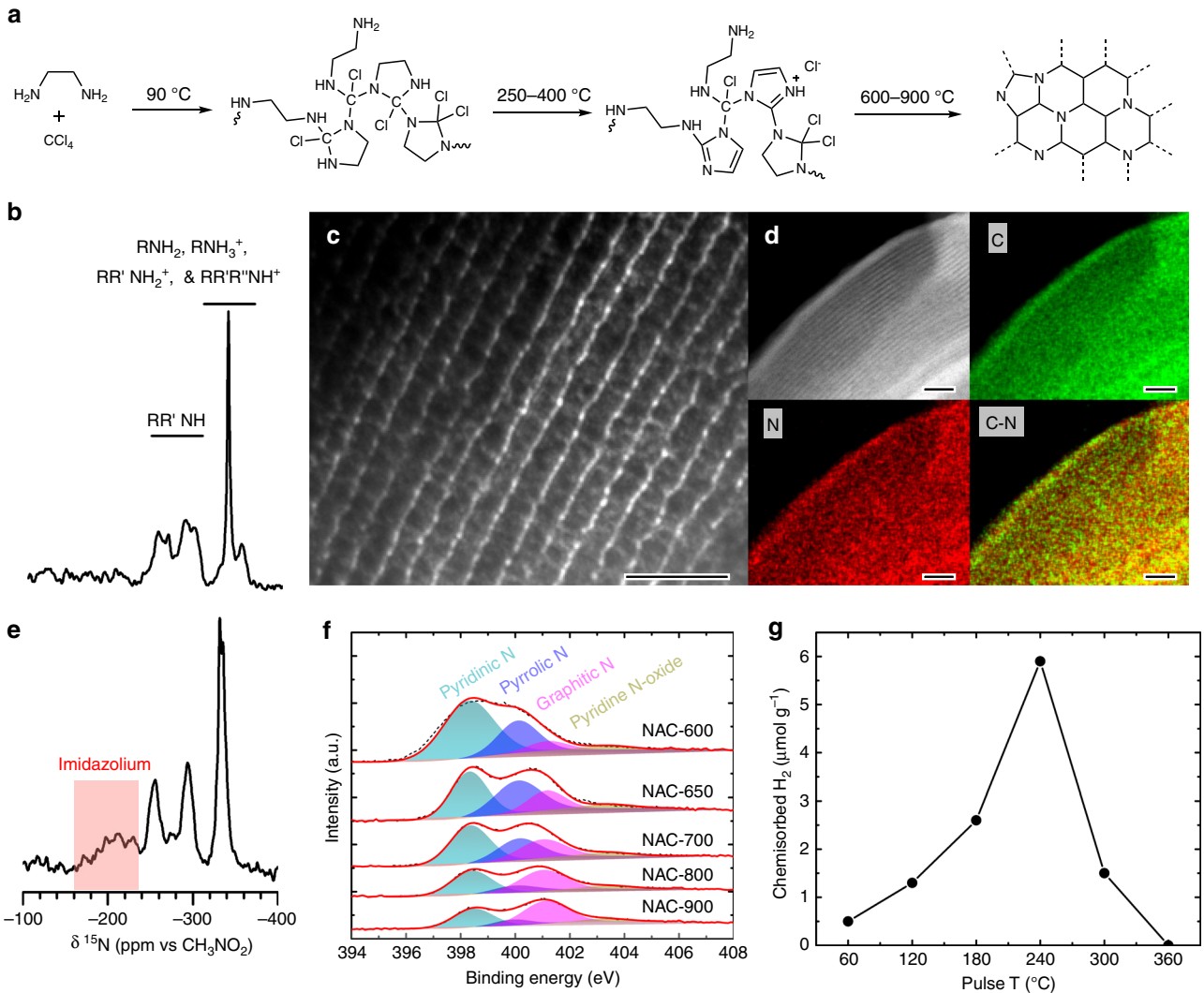

**Fig. 1 Characterization of metal-free carbocatalyst. a** Proposed formation mechanism of NAC catalysts inside the mesoporous silica template. **b** DNP-enhanced $^{15}N\{^1H\}$ CPMAS spectrum of polymer prior to carbonization. **c** TEM image of NAC-800, where 800 represents the sample preparation temperature. Scale bar, 40 nm. **d** HAADF-STEM image and EDS mapping of C and N of NAC-800. Scale bar, 40 nm. **e** DNP-enhanced $^{15}N\{^1H\}$ CPMAS spectrum of condensed polymers calcined for 10 min at 300 °C. **f** N1s XPS spectra of NAC catalysts synthesized at different carbonization temperatures, showing pyridinic N (green), pyrrolic N (blue), graphitic N (magenta), and pyridine N-oxide (brown). **g** Pulsed $H_2$ chemisorption of NAC-800 at 60–360 °C.

reaction (Supplementary Fig. 13). Markedly, the formation of styrene under conditions for hydrogenation and hydrogenolysis is extremely unusual, and contrasts the products observed with common heterogeneous metal catalysts, such as Ni, Pd, and Ru[24,25]. Importantly, these NAC catalysts are the first metal-free catalysts capable of both C–O activation and hydrogenation with $H_2$.

Further kinetic studies with PPE showed that the apparent rate law is zero-order on the PPE and 1.3-order on $H_2$ (Fig. 2c and Supplementary Fig. 14). Hence, under the reaction kinetic regime, the rate-determining step is mainly correlated to the surface dissociation of $H_2$ but not to PPE C–O cleavage. A 1.1-order dependence on the catalyst was also observed, indicating no internal mass transfer limitation[26]. The turn-over frequency (TOF) for PPE conversion can also be estimated, assuming that the density of active sites equals to the maximal $H_2$ uptake quantified by pulsed chemisorption study (at 240 °C, Fig. 1g). Thus, the TOF of NAC-800 for PPE hydrogenolysis (230 °C, 20-bar $H_2$) was calculated to be 32 h$^{-1}$ using initial rates (Supplementary Table 7). Comparison of TOF of catalysts in this work is also made with literature values by metal-based catalysts for similar reactions (Supplementary Table 7).

Strikingly, species resulting from arene hydrogenation of, for example, PPE, 1-phenylethanol, and phenol, were below detection limits. In contrast, supported Pd or Ni catalysts mediate PPE conversions (under the same $H_2$ pressure) to generate significant amounts of hydrogenated arenes as side products (Supplementary Fig. 15). These drastically different selectivities strongly demonstrate that NAC catalysts are advantageous over transition metals in preserving aromaticity of products[27]. Besides, NAC-800 was robust at 230 °C and showed neither obvious deactivation nor changes to selectivities after seven repetitive cycles at ca. 28% PPE conversion (Supplementary Fig. 16). Analysis of recycled NAC-800 indicated no significant changes to catalyst morphology or chemical composition compared with the fresh catalyst (Supplementary Fig. 17 and Supplementary Tables 1, 2, 3, 8, and 9).

The capability in $H_2$ activation also enables NACs as dehydrogenation catalysts in transformation of hydrocarbons. For instance, catalytic non-oxidative dehydrogenation of ethylbenzene over NAC-800 can be achieved to produce styrene in gas phase at 550 °C, affording 20% conversion during a 9-h

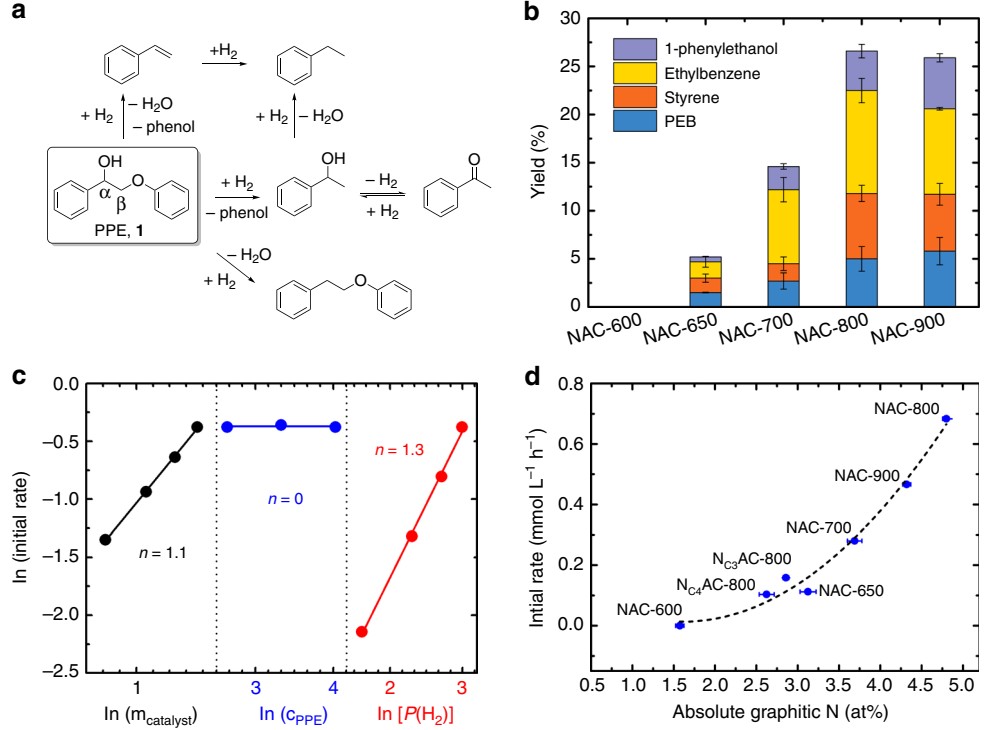

**Fig. 2 Activity results for PPE hydrogenation over the NAC catalysts at 230 °C. a** Reaction network. **b** Yields of products (except phenol). The PPE conversions and yields for 1-phenylethanol (purple), ethylbenzene (yellow), styrene (orange), and PEB (blue) are shown in Supplementary Table 5. Reaction conditions: 14-mmol L$^{-1}$ PPE in 2-PrOH (1.50 mL), NAC catalyst (5.0 mg), 20-bar H$_2$, 8 h. The error bars are standard deviation (s.d.) of repeated tests. **c** Measurement of rate orders for the NAC-800 catalyst (black), PPE (blue), and H$_2$ (red). **d** Plots of initial rates of PPE conversion against the absolute amount of graphitic N in NAC catalysts. The error bars of absolute content of graphitic N were estimated and given in Supplementary Table 3 and the s.d. of the initial rates are ~3% of the mean values, obtained after repeated runs.

time-on-stream test (Supplementary Fig. 18). A possible mechanism (Supplementary Fig. 19) is hypothesized based on an intermediate identified in the mechanistic study with styrene (vide infra). Similarly, the NAC catalyst also demonstrated activity in dehydrogenation of 1,2,3,4-THQ to yield quinoline at 140 °C, and H$_2$ was detected in the headspace (Supplementary Figs. 20 and 21). The dehydrogenation reactions of THQ are much slower under high pressure of H$_2$ (Supplementary Table 10). Under 10-bar H$_2$, a 6% conversion to quinoline was observed at 140 °C in 2 h, while the conversion increased slightly to 10% at 230 °C. The reverse reaction, quinoline hydrogenation, can be also catalyzed by NAC-800, showing a 5% conversion at 140 °C with 10-bar H$_2$. However, increasing the temperature to 230 °C caused the conversion to drop to <1% even with 20-bar H$_2$. Thus, the hydrogenation reaction of quinoline to THQ over NAC-800 is greatly temperature-dependent and can be shut off at higher temperatures.

The NAC-800 catalyst is effective in hydrogenolysis reactions of related compounds with C–O linkages, giving high conversions and selectivities (Fig. 3). The arene-methoxylated PPE derivative, **2**, is converted faster than PPE, delivering guaiacol and 4-ethyl-1,2-dimethoxybenzene in 100% selectivity. Besides, 2-phenoxy-1-phenyl-propane-1,3-diol (PPDE, **3**) is also hydrogenolyzed in 48 h with complete cleavage of aromatic C$_\alpha$–OH and C$_\beta$–O ether, producing propylbenzene, 1-phenyl-propene, and 3-phenylpropanol together with phenol. Quantitative and rapid conversion of arene-methoxylated PPDE derivative, **4**, results in cleavage of all three C–O bonds in 48 h, including the C$_\gamma$–OH. Compound **5**, containing α-O-4 lignin linkage[23], can achieve 92% conversion in 12 h and completion in 16 h, producing toluene and phenol with a TOF of 61 h$^{-1}$ (Supplementary Table 7). Among Compounds **1**–**5** and

1-phenylethanol, C$_\alpha$–O is completely converted; only for **1** and **3**, there is a minor content of uncleaved C$_\beta$–O and C$_\gamma$–O, respectively. The NAC-800 catalyst is also versatile toward hydrogenation of many common unsaturated functionalities[15,28,29], including –C=O, –C=C–, –C≡C–, and –NO$_2$ (Fig. 3), affording the corresponding reduced products with high yields and no arene over-hydrogenation.

**Understanding of surface reactions.** The N assemblies are most likely graphitic because we observed that the initial rates in PPE conversion positively correlate only with absolute graphitic N content of the corresponding NAC catalysts by a second-order polynomial relationship (Fig. 2d). No correlation was apparent with any other nitrogen species (Supplementary Fig. 22). Thus, these N assemblies can be considered as N-heterocycles (e.g., pyrazine and pyrimidine) embedded in graphitic carbon. Such molecular analogs have been employed for hydrogen storage via reversible hydrogenation/dehydrogenation[30,31]. All the NAC materials contain N-pairs, suggested by DNP CPMAS NMR experiments, and therefore we interpret this second-order polynomial correlation as a result of the requirement of assembles of two N-pairs to efficiently activate H$_2$ and subsequent catalysis.

It needs to be noted that carbon black and other nitrogen-doped carbons were found inactive in catalyzing PPE hydro-genolysis, in Supplementary Table 11, including carbon nitride (C$_3$N$_4$) and nitrogen-doped carbon derived from glucose and melamine (N$_m$C$_{glc}$-800) (Supplementary Fig. 23). These results strongly suggest that activation of H$_2$ and C–O is critically dependent upon synergies of closely placed N sites as suggested. The close proximity of N sites in NACs is most likely benefited from the choice of precursors to form imidazolium intermediates.

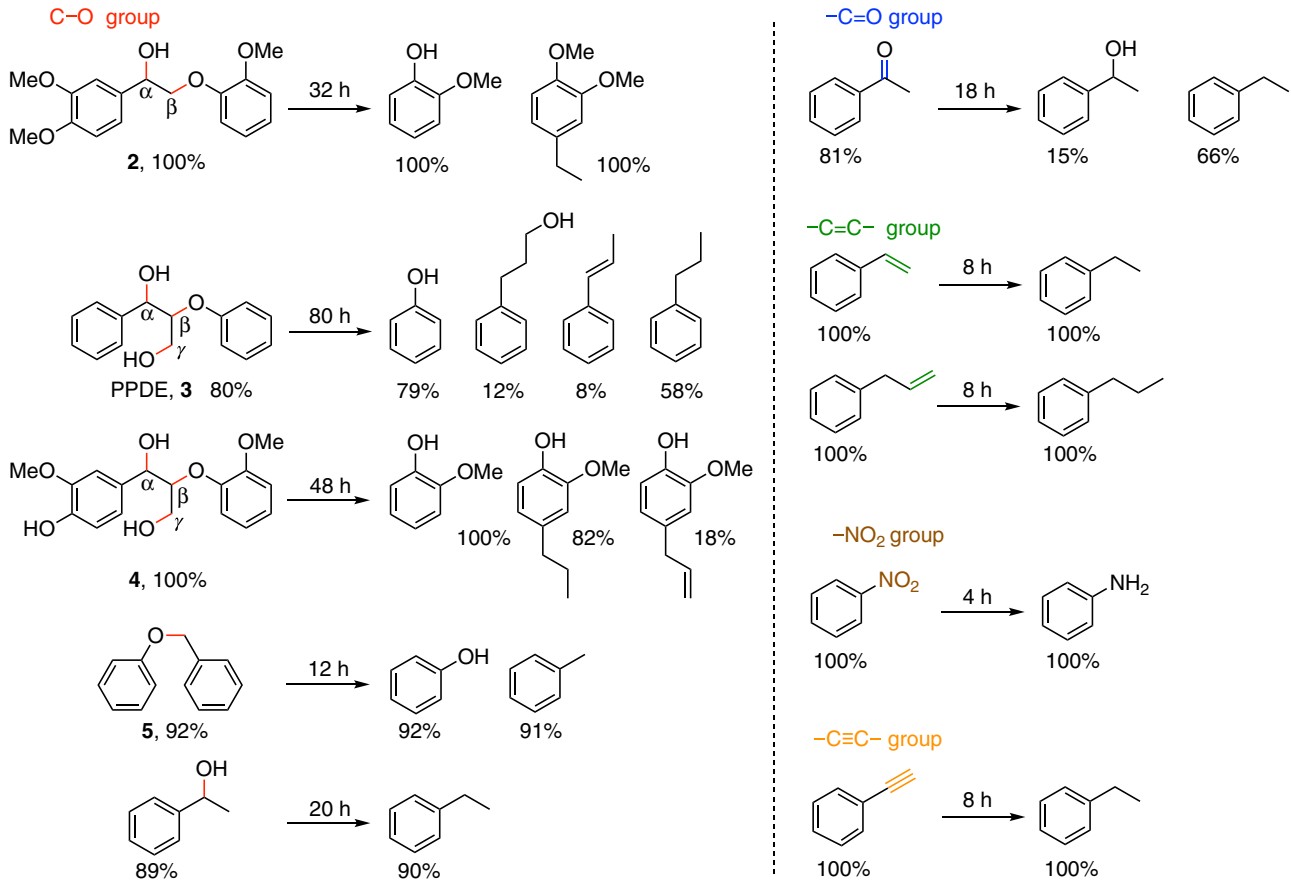

**Fig. 3 Reactions of substrates with different functional groups over the NAC-800 catalyst.** Conversions and molar yields are shown below each substrate and corresponding products, respectively. Reaction conditions: 14-mmo L$^{-1}$ substrate in 2-PrOH (1.50 mL), NAC-800 (5.0 mg), 230 °C, 20-bar H$_2$.

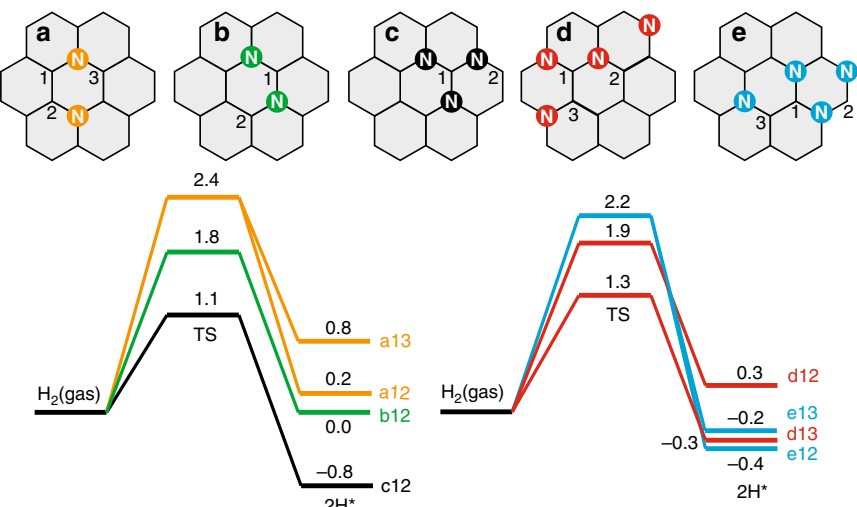

**Fig. 4 DFT calculations of activation of H$_2$ molecules on clustered N sites.** The location of the sites for dissociated hydrogen is labeled by numbers and used in the plot of the energy profile. The values are in eV.

To further understand N assemblies in H$_2$ activation, density functional theory (DFT) calculations using a periodic graphene model were carried out. We find that dissociation of H$_2$ at a single graphitic-N site is endothermic, with an energy cost of 1.4 eV after overcoming an activation barrier of 2.5 eV. This high energy barrier and energy cost suggest isolated graphitic-N are not the kinetically relevant active sites under the reaction conditions. We

then investigated two nitrogen substituted at the para and meta positions (Fig. 4a, b, respectively). On both sites, the dissociative adsorption of H$_2$ has activation barriers of 2.4 and 1.8 eV, respectively. Particularly, around the highly symmetric para N–N dimer, dissociative adsorption of H$_2$ can occur at both a1–a3 and a1–a2 carbon pairs with very similar activation barriers and comparable energy costs. This comparable reaction kinetics for

$H_2$ dissociation could explain the H/D exchange measurement performed and discussed below. That is, $H_2$ and $D_2$ dissociate with similar rates in the proximity (such as a1–a3 and a1–a2 Fig. 4a) of the N–N dimer, and the resulting H or D recombine to form HD.

Furthermore, we find that, when adding one more N in the proximity of the meta N–N pair, this tertiary N cluster has localized charge densities at the carbon in the middle evidenced by the sharp peak at the Fermi level in the density of state calculations (Supplementary Fig. 24). This charge localization leads to a much lower activation barrier of 1.1 eV and a large energy gain of 0.8 eV. The exothermic reaction could cause sluggish sequential hydrogenation or hydrogenolysis reactions because the stable C–H bond has to be broken, which is similar to dissociative $H_2$ adsorption on pyridinic N; the reaction is barrierless, but the large energy gain prohibits further hydrogenation activity (Supplementary Fig. 25).

Intriguingly, when two para N–N clusters are located in the proximity (Fig. 4d, e), we find that the activation barrier could reach 1.3 eV with a moderate energy gain of 0.3 eV. Assuming a pre-exponential factor of $10^{13}$, this leads to a TOF around $2\,s^{-1}$ for hydrogen activation at the reaction temperature. The current calculations thus provide possible configurations for $H_2$ activation at the reaction temperature; however, we cannot exclude other possibilities. The general trend is that the carbon near the paired

substitutional N localizes the electrons, facilitating the H–H dissociation on this metal-free catalyst. Further clustering of substitutional N, such as in the quaternary case, can lower the activation barrier, though such configurations could only occupy a small portion of the total N structures. Approximately 10% N, as depicted in Supplementary Fig. 26, was introduced into a graphene sheet, and we noticed that such clustering of N to form active assembles is likely, which is thermodynamically disfavored without prearrangement of N atoms using specific precursors[32].

The role of N assembly in catalysis is further verified experimentally via the various studies of isotope exchange. The exchange of $H_2$ and $D_2$ (10 bar each) was performed in solvent decane. The isotopically scrambled product HD accumulates over time at 230 °C (Fig. 5a), nearly reaching the equilibrium in 80 h (Fig. 5b). The exchange of $H_2/D_2$ has been demonstrated for defect-rich graphenes in the gas phase[33]. The initial step of the exchange experiment requires the dissociation of $H_2$ and $D_2$ on the surface. The dissociated H/D atoms are covalently bonded to the NAC surface, which are unlikely to diffuse as if they are on transition metals. Considering that decane is incapable of proton shuttling like a protic solvent[34], the detection of HD requires the recombination of H and D chemisorbed on the N assembles of two or more closely placed N sites. We also confirmed deuterium scrambling in decane does not occur with NAC-800 catalyst. Furthermore, the exchange experiment of 20-bar $D_2$ was carried

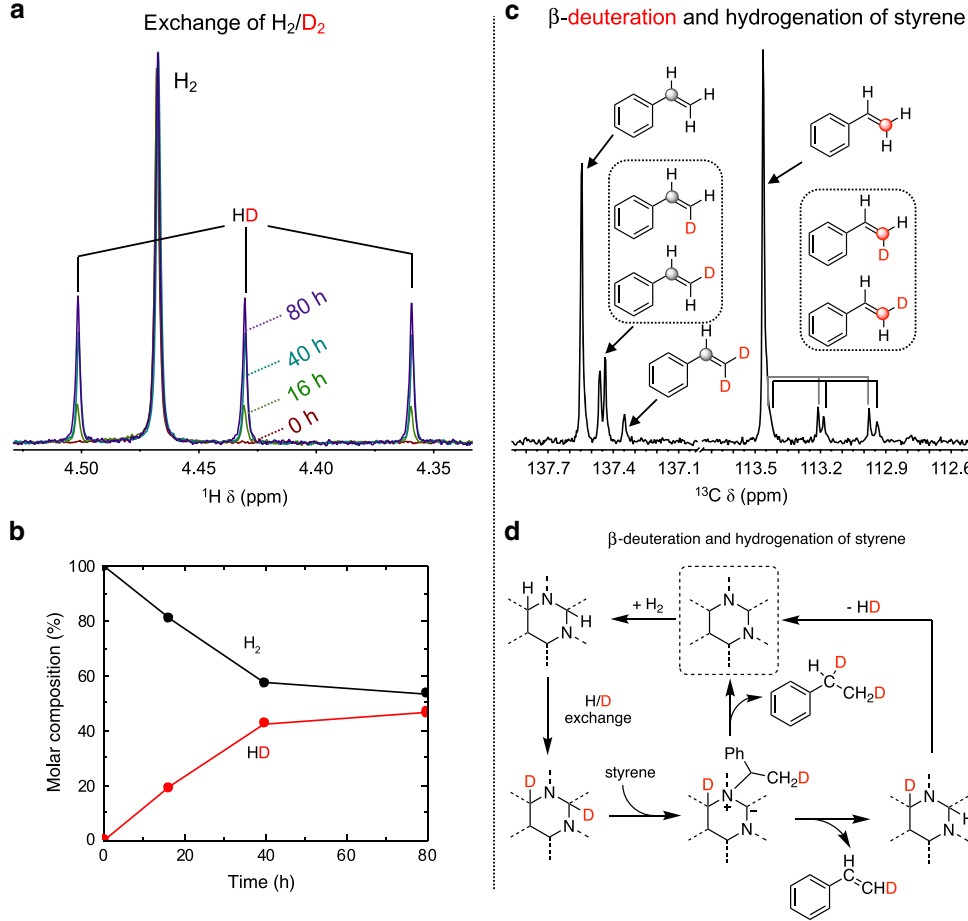

**Fig. 5 Mechanistic studies for the understanding of the active site in NACs. a** $^1H$ NMR spectra dissolved gas from the headspace in benzene-$d_6$ with after $H_2/D_2$ exchange catalyzed by NAC-800, showing the formation of gas HD at 0 (magenta), 16 (green), 40 (cyan), and 80 h (purple). All spectra are normalized to $H_2$ signals. **b** The molar composition of $H_2$ and HD over 80 h. Reaction conditions: NAC-800 (25.0 mg), decane (1.50 mL), 230 °C, $D_2$ (10 bar), and $H_2$ (10 bar). **c** $^{13}C$ NMR spectrum of olefinic carbons in residual styrene after hydrogenation in 2-PrOH-$d_8$ with $H_2$ over NAC-800 (gray and red circles indicate styrene $C_\alpha$ and $C_\beta$, respectively). Reaction conditions: 56-mmol $L^{-1}$ styrene in 2-PrOH-$d_8$ (1.50 mL), 230 °C, 20-bar $H_2$, 4 h. **d** Proposed mechanism of β-selective deuteration and hydrogenation of styrene over NAC-800.

out with solvent 2-PrOH catalyzed by NAC-800 at 230 °C. 2-PrOD-$d_1$, in ca. 10% yield, is detected as a new set of singlets in $^{13}$C NMR in Supplementary Fig. 27, together with the observation of $H_2$ and HD in the headspace (Supplementary Fig. 28), which has only been demonstrated by catalysts based on transition metals like Ni and Pt[35,36]. No D incorporation into the methine of 2-PrOH was observed, suggesting that NAC-800 does not utilize 2-PrOH as a hydrogen source under the reaction condition. To confirm $H_2$ is the sole hydrogen source, we further carried out PPE conversion in 2-PrOH-2-$d_1$ with $H_2$ and detected no formation of HD in the gas phase (Supplementary Fig. 28) or incorporation of D into any reaction intermediates and products (such as PEB and ethylbenzene, Supplementary Fig. 29). The elaborative results of all experiments ($H_2$/$D_2$ and dihydrogen/2-propanol) indicate that the capability of NAC-800 in reversible hydrogen splitting and following exchanges.

To corroborate the participation of N assemblies in catalysis, the reversible H/D exchange experiments are further assessed with the reaction of styrene (an intermediate product in PPE hydrogenolysis). Reactions were carried out in solvent 2-PrOD-$d_8$ with 20-bar $H_2$ at 230 °C. After 4 h, styrene conversion reached 12% with ethylbenzene as the sole product. Up to three deuterium atoms incorporated into the aliphatic carbons of ethylbenzene, as evidenced by the fragments in mass spectra (Supplementary Fig. 30). The deuterium incorporation into the final product, along with the observation of a kinetic isotope effect ($k_H/k_D = 1.8$, Supplementary Fig. 31), indicates that the H/D exchange rate of the active hydrogen with the hydroxyl D in 2-PrOD-$d_8$ is comparable to that of styrene hydrogenation.

Interestingly, residual styrene was also partially deuterated (Supplementary Fig. 30) but selectively at the β-position. Two mono-deuterated and one di-deuterated styrene isotopologues were identified by $^{13}$C solution NMR (Fig. 5c)[37]. However, the experiment with ethylbenzene in a mesitylene solvent does not produce styrene at 230 °C under $N_2$ in the presence of NAC-800, suggesting that direct dehydrogenation of ethylbenzene is not a reverse reaction under such conditions. Therefore, the α-carbon of styrene is selectively chemisorbed to the graphitic N, forming a surface-bound intermediate with a D transferred to the styrene β-carbon (Fig. 5d). The intermediate can be reversibly desorbed by cleaving a $C_β$-H or -D, which leads to the selective β-deuteration of styrene. The same surface intermediate could also be responsible for the direct formation of styrene from PPE.

In summary, we report here that electronically coupled N-assemblies in graphitic carbons can activate $H_2$ molecules and enable selective C–O hydrogenolysis, hydrogenation, and dehydrogenation. Our results establish a new type of active sites (graphitic N assemblies) discovered in the metal-free carbocatalyst, which demonstrates versatile activities only observed among transition metals. This discovery unveils great potential of metal-free carbocatalysts with well-organized surface sites by tuning the molecular precursors and synthetic procedure.

## Methods

**Catalyst preparation**. NAC materials were synthesized based on a modified literature method[17]. SBA-15 (0.80 g) was added into the solution of EDA (1.80 g) and carbon tetrachloride (4.00 g). The condensation of the mixture was carried out at 90 °C for 6 h and dried for 12 h at 120 °C, affording 2.88-g solid residue. The residue was carbonized (3 °C min$^{-1}$, 5 h) at a given temperature under following Ar (MATHESON trigas, 99.999%), resulting in 1.21-g solid. The as-prepared sample was treated with a solution of 5 wt% HF in order to remove Si. The suspension was filtrated, washed with water for ≥10 times and dried under vacuum at 100 °C, rendering 0.32 g of the final sample, named as NAC-$x$ ($x$: calcination temperature). The NAC catalysts were stored in a desiccator under air before usage.

For the synthesis of $C_3N_4$-sheet[38], 10-g melamine was heated in a crucible to 520 °C at a heating rate of 2 °C min$^{-1}$ under air (Airgas, Ultra Zero Grade) and kept for 2 h at 520 °C. After cooling to room temperature, 5-g $C_3N_4$ powder was further heated to 520 °C at a heating rate of 5 °C min$^{-1}$ under air flow (50 mL min$^{-1}$) and

kept for 8 h at 520 °C. After cooling to room temperature again, the sample was heated to 540 °C at a heating rate of 5 °C min$^{-1}$ under $NH_3$ flow (Airgas, 99.995%, 50 mL min$^{-1}$) and kept for 1 h at 540 °C. For the synthesis of $N_mC_{glc}$-800[39], 2-g glucose and 2-g melamine were dissolved in 50-mL deionized water at 60 °C. The solution was then evaporated at 80 °C under air. The powder was transferred into a crucible, heated to 800 °C at a heating rate of 5 °C min$^{-1}$ under a flow of nitrogen and kept for 4 h at 800 °C. For the synthesis of control catalysts with Fe nanoparticles, Fe(acac)$_3$ was impreganated onto NAC-800 by wetness impregnation method. The reduction of Fe was successfully achieved by calcination at 800 °C for 2 h under Ar, named Fe/NAC-800-(800Ar), or by reduction at 500 °C for 2 h in $H_2$, named Fe/NAC-800-(500$H_2$).

**Materials characterization**. Powder XRD was performed on a Bruker D8A25 diffractometer with Cu Kα radiation ($λ = 1.54184$ Å) operating at 30 kV and 25 mA. $N_2$ physisorption was carried out at −196 °C using an auto-adsorption analyzer (Micromeritics, 3Flex). Before adsorption measurements were taken, the samples were degassed at 250 °C overnight. The total pore volume was determined from the aggregation of $N_2$ vapor adsorbed at a relative pressure of 0.99. The specific surface area was calculated using the B.E.T. method, and the pore size was estimated using BJH method from the desorption branch of the isotherms. Transmission electron microscopy images were acquired using a Tecnai G2 F20 electron microscope operated at 200 kV. Aberration-corrected STEM images were recorded using the FEI Titan Themis with an aberration-corrected, mono-chromated, transmission electron microscope operated at 200 kV. Elementary analysis of all samples was carried out using a PerkinElmer 2100 Series II CHN/S Analyzer. The Raman spectra were collected on a Renishaw InVia Raman spectrometer with a microscope accessory. Scanning electron microscopy (SEM) images were acquired on a FEI Quanta 250 FE-SEM. The TGA was acquired on the Netzsch STA449 F1 system equipped with the infrared and mass spectrometer detectors.

In situ DRIFTS study of NAC-800 was carried out using Agilent Cary 670 FTIR equipped with a linearized mercury–cadmium–telluride detector, a Harrick diffuse reflectance accessory, and a Praying Mantis high-temperature reaction chamber. The IR background was collected on KBr packed into the IR sample holder equipped with KBr windows after heating at 400 °C for 2 h under a dynamic He flow (40 mL min$^{-1}$) to remove any adsorbed gaseous molecules. All spectra were obtained at a resolution of 2 cm$^{-1}$ from 1000 to 4000 cm$^{-1}$ under He flow. The DRIFTS spectra of NAC-800 were acquired on a 20× diluted sample with KBr after in situ pretreatment at 400 °C for 2 h under a dynamic He flow (40 mL min$^{-1}$). The in situ DRIFTS spectra were recorded at variable temperatures using the corresponding background collected with KBr at the same temperatures.

The inductively coupled plasma-mass spectroscopy was measured on X Series II, Thermo Scientific to detect the possible metal impurities in the NAC-800. The blank and NAC-800 (10 mg) were separately calcined at 550 °C for 5 h and the residues were completely digested using the hot aqua regia before dilution to 6.0 g with 2.0% nitric acid solution.

XPS were recorded on a PerkinElmer PHI ESCA system by Physical Electronics (PHI) with a detection limit of 1 at%. Monochromatic x-rays were generated by an Al Kα source (1486.6 eV). The binding energy values were strictly calibrated using the C1$s$ peak at 284.6 eV[40]. The N1$s$ peak of assigned N1$s$ species are fitted at an extremely narrow range for the binding energy, including 398.3–398.4 eV for pyridinic N, 400.0–400.2 eV for pyrrolic N, 401.1–401.2 eV for graphitic-N, and 403.3–403.4 eV for nitrogen-oxide. The FWHM values of the fitted peaks are restricted to a range of 1.6–2.0 eV. For quantitative analysis with CASA software, the components were deconvoluted to Gaussian–Lorentzian line shape. The peak area was divided by a sensitivity factor obtained from the element specific Scofield factor and the transmission function of the spectrometer. The uncertainties for deconvolution parameters were estimated using a Monte Carlo approach.

Temperature-programmed desorption (TPD) studies with $CO_2$ or $NH_3$ were carried out on a Micromeritics 3Flex instrument equipped with a mass spec detector. Typically, the NAC-800 catalyst (ca. 100 mg) was pretreated under a flow of He (50 mL min$^{-1}$) at 200 °C for 1 h. The pretreated sample was cooled to room temperature under He (50 mL min$^{-1}$), then $CO_2$ or $NH_3$ (20 mL min$^{-1}$) for 30 min, and finally He (50 mL min$^{-1}$) for 30 min. The TPD data were collected when heating from room temperature to 400 °C at a rate of 10 °C min$^{-1}$ under He flow.

DNP-enhanced $^{15}$N{$^1$H} CPMAS experiments were carried out at 9.4 T on a Bruker 400 MHz DNP solid state NMR spectrometer equipped with a cryo-MAS probe (~−168 °C) and a 263 GHz gyrotron[41]. The samples were exposed to a 16 mM Tekpl solution in 1,1,2,2-tetrachloroethane by incipient wetness in a mortar, and then packed into 3.2 mm sapphire MAS rotors, and spun at 10 kHz.

**Pulsed $H_2$ chemisorption**. Experiments were performed utilizing a Micromeritics AutoChem II instrument under the flow of He at 20 mL min$^{-1}$. NAC-800 catalyst (200 mg) was mounted between quartz wool inside a quartz reactor assembled in a furnace. The temperature was measured at the sample position with a K-type thermocouple sealed in a quartz capillary. All samples were thermally pretreated at 400 °C for 4 h to remove any possible surface contamination such as carbon species or water present in air. In the pulsed chemisorption experiment, $H_2$ consumption was monitored through a thermal conductivity detector that measures the signal difference of the desorbed gas versus a reference flow.

**Catalytic hydrogenolysis and hydrogenation**. The solution (1.50 mL) of the substrate of interest, containing 4.4-mmol L$^{-1}$ dodecane as the internal standard, and NAC catalyst (5.0 mg) were added into a self-built Swagelok reactor (total volume: 5 mL). After purging the reactor with H$_2$ for seven times, the reactor was pressurized with 20-bar H$_2$, and heated to desired temperature under stirring (300 rpm). After the reaction, the reactor was quickly quenched in room-temperature water. In addition to authentic samples, intermediates and products were also verified using GC-MS (Agilent 6890N/5975) with a 30-m capillary column (Agilent, HP-1) and He as the carrier gas. The concentrations of reaction species were quantified using GC-flame ionization detector (FID) (Agilent 6890, FID detector) with a 30-m capillary column (Agilent, HP-1). External calibration was carried out with solutions prepared with authentic samples of related species (Supplementary Fig. 32) and also dodecane. Conversions and yields were further calculated based on the calibration curves established on the dependency of the chromatographic areas as a function of concentration.

The carbon balances were calculated for molecules containing C6 and C8 moieties. For example, in a typical PPE conversion catalyzed by NACs, the C6 carbon balance counts for phenol and PEB together with unconverted PPE; the C8 carbon balance includes styrene, acetophenone, 1-phenylethanol, ethylbenzene, and PEB together with unconverted PPE. The initial rates are determined, using data below 20% conversion. In the recycling study, the catalyst can be recovered by centrifugation and then washed three times with 40-mL 2-PrOH each, followed by drying at 100 °C overnight for next cycle. The uncertainties were estimated based on results from multiple runs.

**Exchange of H$_2$/D$_2$**. Decane (1.50 mL) and NAC-800 catalyst (25.0 mg) were added into a self-built Swagelok reactor (total volume: 5 mL). After purging seven times with H$_2$, the reactor was charged with a total of 20-bar H$_2$ and D$_2$ (1:1). The actual amount of H$_2$ and D$_2$ in the reactor was measured to be at 2.8 and 5.7 mg, respectively. The reactor was then heated to 230 °C under stirring (300 rpm). After different time intervals, the reactor was quickly quenched in room-temperature water. The gas (~2 bar) in the headspace of the reactor was bubbled into 0.5-mL benzene-$d_6$ inside a 5-mm solution NMR tube. $^1$H NMR spectrum of the obtained benzene-$d_6$ with dissolved gas was acquired immediately using Bruker Avance III 600 spectrometer.

**Deuterium incorporation to 2-PrOH**. 1.50 mL of 2-PrOH and NAC-800 catalyst (5.0 mg) was added into a self-built reactor (total volume: 5 mL). After purging three times with D$_2$, the reactor was charged with 20-bar D$_2$, and heated to 230 °C under stirring (300 rpm). After 48 h, the reactor was quickly quenched in room-temperature water. The gas (~2 bar) in the headspace of the reactor was bubbled into 0.5-mL benzene-$d_6$ inside a 5-mm solution NMR tube. $^1$H NMR spectra were measured immediately on the benzene-$d_6$ solution with the dissolved gas using Bruker Avance III 600 spectrometer. $^{13}$C NMR spectra of the final reaction solution was acquired using a co-axial insert with DMSO-$d_6$ using Bruker Avance III 600 spectrometer.

**Styrene hydrogenation**. 56.0-mmol L$^{-1}$ styrene in 2-PrOH or 2-PrOH-$d_8$ (1.50 mL) and NAC-800 catalyst (5.0 mg) was added into a self-built reactor (total volume: 5 mL). After purging seven times with H$_2$, the reactor was charged with 20-bar H$_2$, and heated to 230 °C under stirring (300 rpm). After 2 and 4 h, the reactor was quickly quenched in room-temperature water. $^{13}$C NMR spectrum of the final reaction solution at 4 h was collected on Avance NEO 400 MHz, equipped with liquid N$_2$-cooled broadband Prodigy Probe.

**Catalytic non-oxidative dehydrogenation of ethylbenzene**. The non-oxidative dehydrogenation of ethylbenzene was carried out using a quartz U-tube as a fixed-bed flow reactor under atmospheric pressure. NAC-800 catalyst (20.0 mg) was diluted with 100 mg quartz sand. The temperature of the fixed-bed flow reactor was then raised to 550 °C at a ramping rate of 5 °C min$^{-1}$. At 550 °C, the reaction mixture was passed through the reactor bed. The reaction mixture was composed of 2.5 mL min$^{-1}$ He which was passed through a bubbler containing ethylbenzene (actual ethylbenzene flow = 0.034 mL min$^{-1}$, calculated based on the saturated pressure), 1.0-mL min$^{-1}$ H$_2$ and He as balance gas (total flow = 50 mL min$^{-1}$). The gaseous products from the reactor were monitored online using a HP 5890 gas chromatograph equipped with a capillary column (SE-30, 30 m × 0.32 mm × 0.25 μm) and a FID.

**Catalytic non-oxidative dehydrogenation of THQ**. THQ (0.100 mmol), mesitylene (2.00 mL), and 5.0 mg NAC-800 catalyst were added into a glass reactor attached with an Ar balloon. After different time intervals at 150 °C, the liquid products were quantified using GC-FID (Agilent 6890, FID detector) with a 30-m capillary column (Agilent, HP-1).

For the detection of produced H$_2$, THQ (0.100 mmol), mesitylene (2.00 mL), and 5.0 mg NAC-800 catalyst were added into a self-built Swagelok reactor (total volume: 5 mL). After purging the reactor with N$_2$ for seven times, the reactor was pressurized with 10-bar N$_2$, and heated to 230 °C under stirring (300 rpm). After 3 h, the reactor was quickly quenched in room-temperature water. The liquid products were quantified using GC-FID (Agilent 6890, FID detector) with a 30-m capillary column (Agilent, HP-1). The gas in the headspace of the reactor was bubbled into 0.5-mL benzene-$d_6$ inside a 5-mm solution NMR tube. $^1$H NMR spectrum of the obtained benzene-$d_6$ with dissolved gas was acquired immediately using Bruker Avance III 600 spectrometer.

**Computational simulations**. DFT calculations of hydrogen dissociation on N-doped graphene were performed based on the DFT using the Vienna Ab initio Simulation Package[42]. Exchange and correlation energies were described using the functional proposed by Perdew, Burke and Ernzerhof[43] based on the generalized gradient approximation. The electron-core interactions were treated in the projector augmented wave method[44,45].

The supercell was represented by a slab of one graphene sheet consisting of about 160 carbon atoms and 1–4 nitrogen atoms, which was used in our previous works[46,47]. The x and y length of the supercell were 19.65 and 21.27 Å, respectively, while the vacuum was set to 20 Å, which should be large enough to minimize the interactions between adjacent cells. The atomic structures were fully relaxed for the barrier calculation of H$_2$ dissociation. The kinetic cutoff energy was set to 400 eV. The Brillouin zone was sampled with a single k-point at the G point. The minimum energy pathways were carried out on five models of active sites (Supplementary Fig. 24) using the nudged elastic band (NEB) method[48] and the dimer method[49]. The NEB energy profiles were included in Supplementary Fig. 33. The transition states were further verified by calculating the vibrational frequencies (Supplementary Table 12).

## Data availability

The authors declare that all relevant data supporting the findings of this study are available within the paper and its Supplementary Information files. Additional data are available from the corresponding authors upon reasonable request.

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

## Acknowledgements

Z.L., A.B., T.K., A.S., and L.Q. are supported by the U.S. Department of Energy (DOE), Office of Basic Energy Sciences, Division of Chemical Sciences, Geosciences, and Biosciences. The Ames Laboratory is operated for the U.S. DOE by Iowa State University under Contract No. DE-AC02-07CH11358. We thank the support from Iowa State University. The computational simulations were performed at the OU Supercomputing Center for Education and Research and the National Energy Research Scientific Computing Center (NERSC), a U.S. Department of Energy Office of Science User Facility, and were supported by the U.S. Department of Energy, Basic Energy Sciences (Grant DE-SC0020300). We thank Steven Kmiec and Steve W. Martin for their help with collection of Raman spectra.

## Author contributions

Z.L. and R.N. contributed equally to this work. W.H. and L.Q. designed the experiments. Z.L. and L.Q. performed the liquid-phase experiments and catalyst characterization. V.T.N. and B.W. conducted the DFT calculations. A.B. synthesized several reaction substrates. R.K.B. carried out the gas-phase reactions. X.W. assisted in the catalyst synthesis and characterization. T.K. conducted the DNP experiments. All authors contributed to discussions and participated in data analysis. Z.L., R.N., A.S., B.W., W.H., and L.Q. co-wrote the manuscript with help from all co-authors. B.W., W.H., and L.Q. directed the research.

## Competing interests

The authors declare no competing interests.
