## [Peer Review File · Nature Communications]

Reviewers' comments:

Reviewer #1 (Remarks to the Author):

Ref. NCOMMS-20-07371-T

The work presented in this manuscript discusses results on catalytic cleavage of strong bonds including hydrogen-hydrogen, carbon-oxygen, and carbon-hydrogen over nitrogen-doped graphenes. The main goal for these reactions is the selectivity, but except for the hydrogenation of 2, all others reactions provided either small selectivities (see C-O) or selectivity to less interesting products (N-O) or (C=C). Then, the manuscript provides justifications only for the activation of hydrogen based on H₂/D₂ exchange experiments. The authors assumed the priority of these experiments mentioning that such an exchange “has only been demonstrated by catalysts based on transition metals like Ni and Pt”. However, literature already reported such an exchange for graphene-based materials (G. Sastre, A. Forneli, V. Almasan, V.I Parvulescu, H. Garcia, Isotopic H/D Exchange on Graphenes. A Combined Experimental and Theoretical Study, *Appl. Catal. A: General* 547 (2017) 52-59.).

The investigated reactions also require an interaction of substrate with the catalysts that is not documented. Spectral evidences for such interactions are missing. Calculations’ suggesting that “the α -carbon of styrene is selectively chemisorbed to the graphitic N, forming a surface-bound intermediate with a D transferred to the styrene β -carbon” is important to be experimentally proved. Also important, comparative TOFs should be calculated for the present nitrogen-doped graphene and reference metal catalysts.

Based on these the reviewer considers the manuscript is not enough mature to be published in Nature Communications.

Reviewer #2 (Remarks to the Author):

This manuscript describes development of carbon-based catalyst for hydrogenolysis of C-O linkages in aromatic oxygenates using molecular hydrogen, catalytic dehydrogenation of ethylbenzene and tetrahydroquinoline, and the selective hydrogenation of unsaturated functional groups in aromatics, including $-C=O$, $-C=C-$, $-C\equiv C-$, and $-NO_2$. The carbon-based catalysts were synthesized by condensing ethylenediamine (EDA) and carbon tetrachloride by a hard template route, followed by carbonization under flowing Ar at 600 to 900 °C and then etching to remove the silica template, giving the materials NAC-600-NAC-900. Clearly, utilization of metal-free catalyst for activation of hydrogen at relative mild conditions is very meaningful topic. The results are remarkable but not convincing. This work could be published on Nature Communications if the results can be further confirmed. In addition, there is lack of solid evidence to define the active site and the reaction mechanism is unclear. Here are some specific criticisms:

(1) Fe, Cu and other metals pollution is very common in HF, it is necessary to do more research to see if the presence of trace amounts of Fe, Cu or some other metals would be the real active species. It needs more evidence and control experiments. This is very important to support the conclusion of the current

work.

(2) Some control experiments should be performed to exclude the possibility of 2-PrOH as hydrogen source.

(3) The authors said that “the initial rates in PPE conversion positively correlate only with absolute graphitic N content of the corresponding NAC catalysts by a second-order polynomial.” However, using peak fitting to obtain graphitic N content is not convincing because of overlap of the peaks in N1s XPS spectra.

(4) The mechanisms of dihydrogen dissociation, hydrogenation and dehydrogenation are not clear.

(5) If possible, I know it is difficult, some in-situ characterizations have to be carried to determine the possible active site on the carbon catalyst.

Overall, in view of the above criticisms I do not recommend this paper for publication in the current form. To be suitable for Nature Communications, major revisions are necessary.

Reviewer #3 (Remarks to the Author):

The present study demonstrates that graphitic mesoporous carbon-nitrid materials synthesized following literature recipe can efficiently catalyze hydrogenation, dehydrogenation and hydrodeoxygenation of selected compounds. The synthesis of the catalysts were carefully controlled and the intermediate states and compositions were accurately characterized. N-sites with different oxidation states were identified by XPS (pyridinic, pyrrolic, graphitic and N-O). It has been shown that the catalysts can sufficiently chemisorb H₂. The selected catalyst (NAC-800) could hydrodeoxygenate the model PPE molecule and a considerably high selectivity toward the C-O attack instead of the aryl rings has been shown. This is a clear advantage over many typical TM catalysts. The scope of the hydrodeoxygenation includes C-O, carbonyl, olefinic, alkynyl and NO₂ groups. The same catalyst showed remarkable activity in dehydrogenation and hydrogenation as well. Thorough kinetic analysis has been performed and the kinetic order of the reactant PPE, H₂ and catalyst has been determined. The catalytic mechanism was addressed by the kinetics measurements complemented by periodic DFT calculations. The calculations revealed catalytically feasible route (low TS energetically neutral pathway) and excluded kinetically (high TS) or thermodynamically unfavorable (too stable chemisorption) routes. A remarkable accord in experiment and theory is the plausible explanation of the role of the graphitic N-pairs in the H₂ activation. In my opinion the study provides very new ideas for the metal-free H₂ activation field and also they have disclosed this new type of catalysis for the family of mesoporous carbon-nitrid materials. The study has addressed important catalysis issues (thorough characterizations, different kinetic measurements) and provided convincing mechanistic insights for the operation of the catalysts. I propose the publication of the study. However I have the following issues which need to be addressed:

i) Regarding selectivities: it is shown that NAC-800 boosts orthogonal processes: hydrogenation and dehydrogenation. Hence it would be important/useful to define circumstances or conditions or parameters how to obtain one or the other process during operation.

ii) regarding the calculations: the NEB profiles should be shown in the SI or at least some information

about the imaginary frequencies describing the activation processes probed by the computations should be given.

Reviewers' comments:

Reviewer #1:

The work presented in this manuscript discusses results on catalytic cleavage of strong bonds including hydrogen-hydrogen, carbon-oxygen, and carbon-hydrogen over nitrogen-doped graphenes. The main goal for these reactions is the selectivity, but except for the hydrogenation of 2, all others reactions provided either small selectivities (see C-O) or selectivity to less interesting products (N-O) or (C=C).

Response:

Thanks for the comments. The main goal in this research is to avoid over-hydrogenation of arene, a key issue and a great challenge in traditional transition metal-containing catalyst. The selectivity of the metal-free NAC catalysts is thus defined for conversion of C-O, -NO₂, and C=C *versus* arene hydrogenation. We have stated in the abstract for conversion of C-O that “NACs exhibit high selectivity towards alkylarenes for hydrogenolysis of aryl ethers as model bio-oxygenates without over-hydrogenation of arenes.” For these transformations, transition-metal catalysts are conventionally employed. In the case of C-O hydrogenolysis, catalysts with supported Pd, Pt, and Ni were reported active at 120-150 °C in the presence of 6 - 12 bar H₂ (Zhang, J. et al. *ACS Catal.* **4**, 1574-1583 (2014); He, J. et al. *J. Am. Chem. Soc.* **134**, 20768-20775 (2012)). However, it has been reported in the literature and by our control experiments (Supplementary Fig. 15) that transition metals often lead to the over-hydrogenation of aromatic rings during the catalytic upgrading of phenolic molecules. Such side reactions cause low yields of value-added aromatic products at the cost of H₂ (He, J. et al. *J. Am. Chem. Soc.* **134**, 20768-20775 (2012)). In view of that, NACs are highly desirable as metal-free materials that can solely produce aromatic compounds and prevented over-hydrogenation of phenyl rings. In the substrate scope study (Fig. 3), the numbers beneath the chemical structures are molar conversion and yield for reaction substrates and products, respectively, but not selectivity, as indicated in the figure caption. For the hydrogenolysis of Compounds 1-5 and 1-phenylethanol, C_α-O is completely converted; only for Compounds 1 and 3, there is a minor content of uncleaved C_β-O and C_γ-O, respectively. **Based on the reviewer's comment, we have revised the discussion of the substrate scope study. The revised paragraph is shown below:**

“The NAC-800 catalyst is effective in hydrogenolysis reactions of related compounds with C-O linkages, giving high conversions and selectivities (Fig. 3). The arene-methoxylated PPE derivative, **2**, is converted faster than PPE, delivering guaiacol and 4-ethyl-1,2-dimethoxybenzene in 100% selectivity. Besides, 2-phenoxy-1-phenylpropane-1,3-diol (PPDE, **3**) is also hydrogenolyzed in 48 h with complete cleavage of aromatic C_α-OH and C_β-O ether, producing propylbenzene, 1-phenyl-propene, and 3-phenylpropanol together with phenol. Quantitative and rapid conversion of arene-methoxylated PPDE derivative, **4**, results in cleavage of all three C-O bonds in 48 h, including the C_γ-OH. Compound 5, containing α-O-4 lignin linkage, can achieve 92% conversion in 12 h and completion in 16 h, producing toluene and phenol with a TOF of 61 h⁻¹ (Supplementary Table 7). Among Compounds **1-5** and 1-phenylethanol, C_α-O is completely converted; only for **1** and **3**, there is a minor content of uncleaved C_β-O and

C_γ-O, respectively. The NAC-800 catalyst is also versatile toward hydrogenation of many common unsaturated functionalities²⁸⁻³⁰, including -C=O, -C=C-, -C≡C-, and -NO₂ (Fig. 3), affording the corresponding reduced products with high yields and no arene over-hydrogenation.”

Reduction of both nitrobenzene and olefins are regarded widely as useful organic transformations (Jagadeesh, R. V. *et al. Science* **342**, 1073-1076 (2013); Kong, X. *et al. Energ. Environ. Sci.* **6**, 3260-3266 (2013); Huang, A. *et al. Nat. Commun.* **10**, 10.1038/s41467-019-12460-7, (2019); Jagadeesh, R. V. *et al. Science* **358**, 326-332 (2017)). Although these two reactions can be achieved with transition metals, developing selective and efficient metal-free catalysts permits applications when residues of metal catalysts are detrimental to the application of products. Waymouth and others have repeatedly demonstrated such a strong need to develop new strategies for the use of metal-free catalysts in this area (Zhang, X. *et al. Chem. Rev.* **118**, 839-885 (2018)). The discovery and applications of effective metal-free catalysts based on main group elements can also reduce the consumption of precious metals. **These references have been added to the manuscript.**

(1) Then, the manuscript provides justifications only for the activation of hydrogen based on H₂/D₂ exchange experiments. The authors assumed the priority of these experiments mentioning that such an exchange “has only been demonstrated by catalysts based on transition metals like Ni and Pt”. However, literature already reported such an exchange for graphene-based materials (G. Sastre, A. Forneli, V. Almasan, V.I Parvulescu, H. Garcia, Isotopic H/D Exchange on Graphenes. A Combined Experimental and Theoretical Study, Appl. Catal. A: General 547 (2017) 52-59.).

Response:

We are fully aware of the H₂/D₂ exchange work by Garcia group and have cited their first publication (Primo, A. *et al. Nat. Commun.* **5**, 5291 (2014), <https://doi.org/10.1038/ncomms6291>) in our initially submitted manuscript. **The paper mentioned by the reviewer and the following sentence are now added into the revised manuscript.**

“The exchange of H₂/D₂ has been demonstrated for defect-rich graphenes in the gas phase³⁴.”

Our study includes the exchange of surface chemisorbed hydride with not only D₂ but also exchangeable proton of solvent 2-propanol. We have demonstrated the exchange of 2-PrOH + D₂ and 2-PrOD-*d*₈ + H₂. Such approach has been reported for catalysts based on transition metals by Whitesides (Pt at -20 and 40 °C, Lee, T. *et al. J. Am. Chem. Soc.* **113**, 2568-2576 (1991)) and ourselves (Ni/Al₂O₃ at 175 °C, Qi, L. *et al. J. Am. Chem. Soc.* **141**, 17370-17381 (2019)). In addition, to answer one comment of the 2nd reviewer, we also carried out the exchange study of 2-PrOH-2-*d*₁ + H₂ in the presence of PPE. As expected, we did not detect any H/D exchange or deuterium incorporation into the final products, which exclude 2-PrOH as the hydrogen source. All these exchange experiments suggest that only the hydroxyl proton of 2-PrOH is exchangeable with surface chemisorbed hydrogen under the reaction conditions. Hence, on the

basis of reviewer's suggestion, we have revised the respective sentences in the manuscript to the following:

“Furthermore, the exchange experiment of 20 bar D₂ was carried out with solvent 2-PrOH catalyzed by NAC-800 at 230 °C. 2-PrOD-*d*₁, in ca. 10% yield, is detected as a new set of singlets in ¹³C NMR in Supplementary Fig. 28, together with the observation of H₂ and HD in the headspace (Supplementary Fig. 29), which has only been demonstrated by catalysts based on transition metals like Ni and Pt^{36,37}. No D incorporation into the methine of 2-PrOH was observed, suggesting that NAC-800 does not utilize 2-PrOH as a hydrogen source under the reaction condition. To confirm H₂ is the sole hydrogen source, we further carried out PPE conversion in 2-PrOH-2-*d*₁ with H₂ and detected no formation of HD in the gas phase (Supplementary Fig. 29) or incorporation of D into any reaction intermediates and products (such as PEB and ethylbenzene, Supplementary Fig. 30). The elaborative results of all experiments (H₂/D₂ and dihydrogen/2-propanol) indicate that the capability of NAC-800 in reversible hydrogen splitting and following exchanges.”

(2) The investigated reactions also require an interaction of substrate with the catalysts that is not documented. Spectral evidences for such interactions are missing. Calculations' suggesting that “the α-carbon of styrene is selectively chemisorbed to the graphitic N, forming a surface-bound intermediate with a D transferred to the styrene β-carbon” is important to be experimentally proved.

Response:

We feel that our original statement has caused a misunderstanding. Rather than based on theoretical calculations as the reviewer thought, the specific interaction of styrene with the surface is indeed based on spectroscopic studies (NMR and MS) of reaction mixtures before and after reactions.

Hydrogenation of styrene in 2-PrOD-*d*₈ with H₂ was performed over NAC-800, and we discovered that some non-reduced styrene was selectively deuterated at the β-position (Fig. 5c and Supplementary Fig. 31). The NMR identification of styrene deuteration has been further confirmed by investigating high-resolution MS fragments. The experiment with ethylbenzene does not produce styrene at 230 °C under N₂ in the presence of NAC-800. Therefore, direct dehydrogenation of ethylbenzene is not reversible under such reaction conditions and the deuteration is not caused by C-H activation of ethylbenzene.

Based on the experimental evidence, the most reasonable mechanism is deduced and shown in Fig. 5d. The surface-bound intermediate is responsible for the formation of styrene β-deuteration after the α-carbon of styrene is selectively chemisorbed to the graphitic N when a deuterium atom is transferred to the styrene β-carbon. The sorption to surface carbon atoms will generate chemically inert C-C bonds that cannot be cleaved easily to complete the catalytic cycle. Similar rationale to deduce the proposed mechanism can be found in a recent report (Puleo, T. R. et al *J.*

Am. Chem. Soc. **141**, 1467-1472 (2019)) for α -deuteration of styrenic substrates by a homogeneous catalyst, which is cited in our initially-submitted manuscript.

We also revised the manuscript to clarify that the identification of surface species responsible for selectively β -deuteration of styrene is based on experimental evidence but not by calculations. The revised paragraph is shown below:

“To corroborate the participation of N assemblies in catalysis, the reversible H/D exchange experiments are further assessed with the reaction of styrene (an intermediate product in PPE hydrogenolysis).”

(3) Also important, comparative TOFs should be calculated for the present nitrogen-doped graphene and reference metal catalysts.

Response:

TOF calculations are very challenging to obtain for metal-free catalysts and thus not usually reported (Grant, J. et al. *Science* **354**, 1570-1573 (2016); Zhang, J. et al. *Science*, **322**, 73-77 (2008)). We estimate the TOF using the typical method for catalysts based on transition metals, assuming that the density of surface active sites equals to the H₂ uptake quantified by pulsed chemisorption study. The highest H₂ sorption is 5.9 $\mu\text{mol g}^{-1}$, measured at 240 °C. Thus, the TOF values for the conversion of PPE and benzyl phenyl ether (**5**, BPE) by NAC-800 were calculated to be 32 and 61 h⁻¹, respectively, based on initial rates. Our TOF values for BPE and PPE conversions are compared to several representative metal-based catalysts in literature. Both our results and literature values are now summarized in Supplementary Table 7. The following sentences are also added in the manuscript:

“The turn-over frequency (TOF) for PPE conversion can also be estimated, assuming that the density of active sites equals to the maximal H₂ uptake quantified by pulsed chemisorption study (at 240 °C, Fig. 1g). Thus, the TOF of NAC-800 for PPE hydrogenolysis (230 °C, 20 bar H₂) was calculated to be 32 h⁻¹ using initial rates (Supplementary Table 7). Comparison of TOF of catalysts in this work is also made with literature values by metal-based catalysts for similar reactions, Supplementary Table 7.”

“Compound **5**, containing α -O-4 lignin linkage, can achieve 92% conversion in 12 h and completion in 16 h, producing toluene and phenol with a TOF of 61 h⁻¹ (Supplementary Table 7).”

Supplementary Table 7 Calculated TOF values of NAC-800 and other reference metal catalysts.

catalyst	reactant	temperature (°C)	H ₂ (bar)	time (h)	Conversion (%)	TOF (h ⁻¹)	Reference
NAC-800 ^a	1 , PPE	230	20	2	8.9	32	This work
NAC-800 ^a	5 , BPE	230	20	2	17	61	This work
N _m C _{glc} -800	1 , PPE	230	20	8	0	0	This work
C ₃ N ₄	1 , PPE	230	20	8	0	0	This work
57 wt% Ni/SiO ₂	PEB	120	6	1.5	8.1	20	J. Am. Chem. Soc. 134 , 20768-20775 (2012)

5 wt% Ru/C	1, PPE	150	-- ^b	10	66.7	1.3	ACS Sus. Chem. Eng. 6 , 2872-2877 (2018)
56 wt% Ni/Al ₂ O ₃	5, BPE	130	-- ^b	-	15.4	0.27	Chem. Sci. 10 , 4458-4468 (2019)

^a the density of active sites is assumed to be the same as the hydrogen uptake measured and calculated by pulsed chemisorption at 240 °C. ^b using 2-PrOH as hydrogen source.

Based on these the reviewer considers the manuscript is not enough mature to be published in Nature Communications.

Response:

We beg to disagree with the reviewer, and believe that the novelty of this work is well justified and extensive data have been collected to support it as follows. We have clarified the desired selectivity in C-O hydrogenolysis chemistry and restated the innovation of catalysts and the corresponding application, in both manuscripts and the response to the questions. The chemisorption of H₂ at reaction temperatures and the surface exchange of chemisorbed hydrogen with exchangeable hydrogen in the solvent molecules are elucidated in multiple experimental studies, also supported by theoretical evidence.

We also gained valuable information on the surface sorption and reactions using probe reactions with styrene (also a unique intermediate in the PPE reaction). Similar reactions were used in recent reports for single-site catalysts (Liu, P. et al. *Science*, **352**, 797-800 (2016)) and for homogeneous catalysts (Puleo, T. R. et al. *J. Am. Chem. Soc.* **141**, 1467-1472 (2019)). The intermediate deduced on styrene is also highly likely to be the key surface species in PPE reactions and ethylbenzene dehydrogenation.

Therefore, the main strength of the NAC catalysts is the versatility in the activation of strong bonds (including H-H, C-O, and C-H) and other transformations. These organic reactions are then strongly linked to the N-assemblies as active sites, investigated and described with a broad range of probe reactions, control reactions, kinetic studies, and advanced spectroscopic characterization. We believe this work is of general interests to a broad range of researchers and will provide valuable guidance for developing metal-free materials with unique catalytic performance in the future.

Reviewer #2:

This manuscript describes development of carbon-based catalyst for hydrogenolysis of C-O linkages in aromatic oxygenates using molecular hydrogen, catalytic dehydrogenation of ethylbenzene and tetrahydroquinoline, and the selective hydrogenation of unsaturated functional groups in aromatics, including $-C=O$, $-C=C-$, $-C\equiv C-$, and $-NO_2$. The carbon-based catalysts were synthesized by condensing ethylenediamine (EDA) and carbon tetrachloride by a hard template route, followed by carbonization under flowing Ar at 600 to 900 °C and then etching to remove the silica template, giving the materials NAC-600-NAC-900. Clearly, utilization of metal-free catalyst for activation of hydrogen at relative mild conditions is very meaningful topic. The results are remarkable but not convincing. This work could be published on Nature Communications if the results can be further confirmed. In addition, there is lack of solid evidence to define the active site and the reaction mechanism is unclear.

(1) Fe, Cu and other metals pollution is very common in HF, it is necessary to do more research to see if the presence of trace amounts of Fe, Cu or some other metals would be the real active species. It needs more evidence and control experiments. This is very important to support the conclusion of the current work.

Response:

We have considered this issue and implemented extensive measures to exclude the contamination problems before our initial submission. In addition, more control experiments were also conducted on the basis of the valuable comments. Below is a detailed summary of evidences and control experiments to support this conclusion.

First, we have shown that the PPE reaction network by NAC catalysts with the formation of several intermediates, including styrene, 1-phenylethanol, acetophenone, and PEB, utterly different from those by Pd/Al₂O₃ and Ni/C already included in the initially submitted work (Supplementary Fig. S15). The observed β -deuteration of styrene further supports the new type of active sites other than metals. The mechanistic evidence strongly indicates the NAC offers a unique type of site and reaction mechanism from metal sites. It has been indicated in the initially submitted manuscript that “the formation of styrene under conditions for hydrogenation and hydrogenolysis is extremely unusual, and contrasts the products observed with common heterogeneous metal catalysts, such as Ni, Pd, and Ru.”

Second, all the acids used in this study, including HF, HNO₃, and HCl, are of TraceMetal Grade (Fisher Scientific, contains <1 ppb for Co, Cu, Fe, Ni, Ru, Pd, Pt, Rh, Al, Sb, As, Ba, Be, Bi, B, Cd, Ca, Ce, Cs, Cr, Dy, Er, Eu, Gd, Ga, Ge, Au, Hf, Ho, In, La, Pb, Li, Lu, Mg, Mn, Hg, Mo, Nd, Nb, K, Pr, Re, Rb, Sm, Sc, Se, Ag, Na, Sr, Ta, Te, Tb, Tl, Th, Tm, Sn, Ti, W, U, V, Yb, Y, Zn, and Zr). **The purity information of the inorganic acids is updated in the experimental session.** Besides, we synthesized three different batches of NAC-800, using two sources of carbon tetrachloride and two sources of ethylenediamine (Supplementary Table 3). The resulting NAC-800 catalysts show identical activities for PPE conversion after repeated tests. These results

imply that actual active centers from the NAC-800 are not caused by inorganic and organic impurities that may originate from the chemicals used in this study.

Third, we have carried out ICP-MS analysis of metal-free NAC-800 samples. The detection limit for the ICP-MS is 1 ppb, allowing the detection and quantification of metal loadings as low as 0.00006 wt%. The 5-point calibrations (other than single-point calibrations) were achieved for seven common transition metals (Cu, Co, Ni, Ru, Pd, Pt, and Rh) used for hydrogenation and hydrogenolysis. In the sample prepared with NAC-800, the measured concentrations of these metals are below the detection limit. This result indicates that there are no appreciable amounts of metal impurities which may serve as the actual active sites of our catalysts. **We updated in the manuscript with the following statement:**

“signals for other elements, particularly metals (e.g. Cu, Co, Ni, Ru, Pd, Pt, and Rh), were below detection limits of ICP-MS, STEM-EDX, and XPS.”

Last, we performed five control experiments to demonstrate metal salts and supported iron particles are ineffective. FeCl₃, NiCl₂, or CuCl₂ were added in the reaction solution and showed no activity in PPE conversion. We also synthesized control catalysts with Fe nanoparticles by wetness impregnation of Fe(acac)₃ onto NAC-800. The reduction of Fe was successfully achieved by calcination at 800 °C for 2 h under Ar, named Fe/NAC-800-(800Ar), or by reduction at 500 °C for 2 h in H₂, named Fe/NAC-800-(500H₂). The corresponding Fe loadings are 7.5 and 0.7 wt%, respectively. Both supported Fe catalysts on NAC-800 demonstrate comparable or lower activities in comparison with metal-free NAC-800 for the PPE conversion. **Results of the control experiments are now added in the revised manuscript as Supplementary Table 6.**

“No PPE conversion was observed in the control experiments with control catalysts containing (such as Fe, Cu, Ni in Supplementary Table 6).”

Supplementary Table 6 PPE conversion by various control catalysts. Reaction conditions: control catalyst, 14 mmol L⁻¹ PPE in 2-PrOH (1.50 mL), H₂ (20 bar), 230 °C, 4 h.

exp	catalysts	conversion (%)	yield (%)						carbon balance (%)	
			a	b	c	d	e	f	C6	C8
1	NAC-800	19	16	3	13	0	3	0	100	100
2	FeCl ₃ ^a	0	-	-	-	-	-	-	-	-
3	CuCl ₂ ^a	0	-	-	-	-	-	-	-	-
4	NiCl ₂ ^a	0	-	-	-	-	-	-	-	-
5	Fe/NAC-800-(800Ar) ^b	14	9	1	0	0	5	7	100	99
6	Fe/NAC-800-(500H ₂) ^c	19	14	3	1	0	5	9	100	99

^a 0.105 mmol; ^b 5.0 mg, Fe loading is 7.5 wt%; ^c 5.0 mg, Fe loading is 0.7 wt%.

(2) Some control experiments should be performed to exclude the possibility of 2-PrOH as hydrogen source.

Response:

Previously the exchange experiment of 20 bar D_2 was carried out with solvent 2-PrOH catalyzed by NAC-800 at 230 °C. 2-PrOD- d_1 is detected in Supplementary Fig. 28, together with the observation of H_2 and HD in the headspace (Supplementary Fig. 29). No D incorporation into the methine of 2-PrOH was observed in the presence of NAC-800, suggesting that 2-PrOH does not undergo reversible dehydrogenation/hydrogenation and 2-PrOH is not a hydrogen source under the reaction condition.

Furthermore, based on this reviewer's suggestion, we performed the isotopic exchange experiment with the solvent 2-PrOH-2- d_1 and 20 bar H_2 at 230 °C catalyzed by NAC-800. No HD in the headspace (Supplementary Fig. 29) was detected after 48 h reaction, and no D incorporation into intermediates and products was detected (Supplementary Fig. 30). Both results indicate that NAC-800 does not activate 2-PrOH as a hydrogen source under the reaction condition (20 bar H_2). In the revised manuscript, these supporting results are integrated into the existing Supplementary Figs. 29 and 30. The corresponding descriptions were added in the manuscript:

“To confirm H_2 is the sole hydrogen source, we further carried out PPE conversion in 2-PrOH-2- d_1 with H_2 and detected no formation of HD in the gas phase (Supplementary Fig. 29) or incorporation of D into any reaction intermediates and products (such as PEB and ethylbenzene, Supplementary Fig. 30).”

Supplementary Fig. 29 1H NMR spectra of the benzene- d_6 with dissolved gas: (a) pure D_2 , showing no residual H_2 or HD was found in the starting D_2 ; (b) the headspace after the exchange reaction of 2-PrOH with D_2 , showing the formation of HD and H_2 ; and (c) the headspace after the exchange reaction of 2-

PrOH- d_1 with H_2 in the presence of PPE (14 mmol L^{-1}), showing only H_2 . Reaction conditions: NAC-800 catalyst (5.0 mg), corresponding solvent (1.50 mL), $230 \text{ }^\circ\text{C}$, 48 h , hydrogen (20 bar).

Supplementary Fig. 30 Mass spectra of product (a) ethylbenzene and (b) PEB in 2-PrOH- d_1 over NAC-800 catalyst, showing no deuterium incorporation. Reaction conditions: NAC-800 (5.0 mg), 14 mmol L^{-1} PPE in 2-PrOH- d_1 (1.50 mL), $230 \text{ }^\circ\text{C}$, 48 h , H_2 (20 bar).

(3) The authors said that “the initial rates in PPE conversion positively correlate only with absolute graphitic N content of the corresponding NAC catalysts by a second-order polynomial.” However, using peak fitting to obtain graphitic N content is not convincing because of overlap of the peaks in N1s XPS spectra.

Response:

We note that, although the pyridinic N, pyrrolic N, graphitic-N, and pyridine N-oxide species are overlapped in N1s XPS spectra, reasonable and reliable peak fitting can be achieved by following two rules. One is that the peak position, shape, and other parameters for the same N specie in different NACs must be consistent; the other is that different N species in a NAC must follow the theoretical values of N1s core level energy shifts (Ayiania, M. et al. *Carbon*, **162**, 528-544 (2020)). In fact, successful fitting of N1s XPS spectra has been demonstrated in many well-accepted applications to understand active sites and surface reactions (Westerhaus, F. et al. *Nat. Chem.* **5**, 537–543 (2013); Guo, D. et al. *Science*, **351**, 361-365 (2016); Cui, X. et al. *Nat. Commun.* **7**, 1-8 (2016); Varnell, J. et al. *Nat. Commun.* **7**, 12582-12590 (2016); Jagadeesh, R. et al. *Science*, **358**, 326–332 (2017)).

To reassure the accuracy of XPS deconvolution, we performed deconvolution of the N1s peak in NACs again by exactly obeying the following stringent parameters:

- (1) The N1s peak of assigned N1s species are fitted with an extremely narrow range for the binding energy, including $398.3\text{-}398.4 \text{ eV}$ for pyridinic N, $400.0\text{-}400.2 \text{ eV}$ for pyrrolic N, $401.1\text{-}401.2 \text{ eV}$ for graphitic-N, and $403.3\text{-}403.4 \text{ eV}$ for nitrogen-oxide (Ayiania, M. et al. *Carbon*, **162**, 528-544 (2020)).

- (2) The FWHM values of the fitted peaks are restricted to a range of 1.6-2.0 eV. (Ayiania, M. et al. *Carbon*, **162**, 528-544 (2020))
- (3) All N1s spectra were strictly calibrated by the C1s peak located at 284.6 eV according to the latest reference (Greczynski, G. et al. *Angew. Chem. Int. Ed.* **59**, 5002-5006 (2020)).

The new fitting parameters and simulated data were shown in Supplementary Table 2 and Fig. 1f, respectively. The changes in relative contents for the four N species in the new data (Supplementary Table 3) are less than 10 % from the previous fitting. With the new data (Supplementary Fig. 9), the initial rates in PPE conversion still positively correlate only with the absolute graphitic N content of the corresponding NAC catalysts in a second-order polynomial relationship (Fig. 2d). More importantly, no specific correlations of catalytic activities can be found with other N species (Supplementary Fig. 22).

In the revised manuscript, these fitting criteria have been incorporated into the experimental session. Figs. 1f and 2d, Supplementary Fig. 9, and Supplementary Table 3 are updated. Supplementary Fig. 22 and Supplementary Table 2 are added in supplementary information. The following sentence is added in the manuscript:

“No correlation was apparent with any other nitrogen species (Supplementary Fig. 22).”

Supplementary Table 2 N1s fitting parameters for NACs. The N 1s peak of assigned N species are fitted with a narrow range for the binding energy according to theoretical calculation.⁶

catalyst	pyridinic N		pyrrolic N		graphitic N		pyridine N-oxide	
	position	FWHM	position	FWHM	position	FWHM	position	FWHM
NAC-600	398.3	2.0	400.1	1.7	401.2	1.7	403.4	2.0
NAC-650	398.3	1.6	400.1	2.0	401.1	1.6	403.4	2.0
NAC-700	398.3	1.7	400.1	1.8	401.0	2.0	403.4	2.0
NAC-800	398.4	1.8	400.0	1.9	401.0	2.0	403.4	2.0
NAC-900	398.4	1.7	400.0	1.6	401.1	1.9	403.3	1.8
N _{C3} AC-800	398.4	1.8	400.1	1.5	401.0	2.0	403.4	1.9
N _{C4} AC-800	398.4	1.7	400.1	1.6	401.0	2.0	403.4	1.6

Fig. 1f N1s XPS spectra of NAC catalysts synthesized at different carbonization temperatures.

Supplementary Table 3 N1s fitting data of NACs.

entry	catalysts	pyridinic N	pyrrolic N	graphitic N	pyridine N-oxide
1	NAC-600	61.1	27.0	8.0	3.9
2	NAC-650	41.1	36.5	17.9	4.5
3	NAC-700	44.3	26.3	24.6	4.7
4	NAC-800	40.8	12.9	40.1	6.2
5	NAC-900	35.5	10.7	48.0	5.7
6	N _{C3} AC-800	35.2	10.1	49.1	5.6
7	N _{C4} AC-800	34.2	10.8	49.9	5.0

Supplementary Fig. 9 The absolute contents of pyridinic, pyrrolic, graphitic, and pyridine N-oxide nitrogen species in NACs by deconvolution of XP spectra. Higher calcination temperature leads to the decrease of both pyridinic and pyrrolic N, while graphitic N increases with temperature first at 600-800 °C and then drop at 900 °C. The total N contents of NACs synthesized with longer-chain diamines are lower.

Fig. 2d Plots of initial rates of PPE conversion against the absolute amount of graphitic N in NAC catalysts.

Supplementary Fig. 22 The correlation between absolute contents of each N species of different NAC catalysts and corresponding initial rates of PPE conversions. No correlation of initial rates was found with N species except for graphitic N (Fig. 2d).

(4) *The mechanisms of dihydrogen dissociation, hydrogenation and dehydrogenation are not clear.*

Response:

For the dihydrogen dissociation, the DFT calculations suggest that the N dopants activate the neighboring carbon atoms, as shown by the electronic states at the Fermi level in the DOS plots in Supplementary Fig. 24. The activated carbon atoms, which are otherwise chemically inert in the absence of nitrogen, thus catalyze the H₂ dissociation, forming two C-H bonds. For example, DFT calculations in the manuscript showed that the dissociative adsorption of H₂ on NAC could occur at a1-a2 carbon pairs with very similar activation barriers and comparable energy costs (Fig. 4 and Supplementary Fig. 24). This comparable reaction kinetics for H₂ dissociation could explain the H/D exchange, that is, H₂ and D₂ dissociate with similar rates in the proximity of the N-N dimer, and the resulting H or D recombine to form HD. **We have revised the following sentence in the text:**

“That is, H₂ and D₂ dissociate with similar rates in the proximity (such as a1-a3 and a1-a2 Figure 4a) of the N-N dimer, and the resulting H or D recombine to form HD.”

For hydrogenation, solid evidence has been discovered in this work by performing isotope labeling studies for styrene hydrogenation. We carried out hydrogenation of styrene in 2-PrOD-*d*₈ solvent with 20 bar H₂ at 230 °C. Besides ethylbenzene formation, the β-carbon of unconverted styrene was found to be deuterated while α-carbon remains untouched (Fig. 5c). The selective β-deuteration can only be achieved when the styrene is chemisorbed on the catalyst with α-carbon bonded to the graphitic N and a surface-bound D transferred to the styrene β-carbon (Fig. 5d). The intermediate can be reversibly desorbed by cleaving a C_β-H or -D, which leads to the selective β-deuteration of styrene.

Fig. 5d Proposed mechanism of hydrogenation of styrene over NAC-800.

For dehydrogenation, it is reasonable to believe that the same intermediate in styrene hydrogenation can be formed after ethylbenzene is sorbed. The formation of styrene can be achieved when the leaving H is transferred onto the catalyst surface, following the reversed hydrogenation pathway. In the revised manuscript, Supplementary Fig. 19 has been added in supplementary information and the corresponding description was added into the manuscript:

“A possible mechanism (Supplementary Fig. 19) is hypothesized based on an intermediate identified in the mechanistic study with styrene (*vide infra*).”

Supplementary Fig. 19 Proposed mechanism of dehydrogenation of ethylbenzene to styrene over NAC-800.

(5) *If possible, I know it is difficult, some in-situ characterizations have to be carried to determine the possible active site on the carbon catalyst.*

Response:

As the reviewer noted, the current challenge in the research area of metal-free catalysts is to determine the active sites and surface reactions *via in situ* methods. For example, STEM can reveal the hetero-atoms sites of 2-D materials like graphene but is less useful for 3D catalysts (Hofer, C. et al. *Nat Commun* **10**, 4570 (2019)). To gain such information, we have conducted *in situ* DRIFTS and *ex situ* solid-state NMR. The results of *in situ* DRIFTS clearly shows the C-H formation at 240 °C.

The *in situ* DRIFTS experiments of NAC-800 were conducted under the flow of He and H₂. The NAC-800 catalyst was diluted 20× with KBr due to the strong adsorption of IR irradiation by the black NAC-800 powder. After *in situ* pretreatment at 400 °C for 2 h under a dynamic He flow (40 mL/min), the DRIFTS spectra were collected at variable temperatures. Under the flow of He at 30 °C, only graphitic *sp*² domains (1578-1593 cm⁻¹) and C–N bond (1269-1290 cm⁻¹) can be observed (Supplementary Fig. 6a). The *in situ* experiments of NAC-800 were carried out up to 240 °C, at which we detected the largest amount of chemisorbed hydrogen on the NAC-800 by the H₂ pulsed chemisorption (Fig. 1g). The spectrum of NAC-800 did not change at 240 °C under He, which was used as background for later studies (Supplementary Fig. 6b). Then, the atmosphere was switched from He to H₂ and changes to the spectra were observed in the range of 4000-2000 cm⁻¹ (Supplementary Figs. 6c and 6d). A new peak showed up at ~2930 cm⁻¹, ascribed to C–H stretching vibration (Johnson *et al. Nat. Commun.* **6**, 7628 (2015)). The low intensity is due to necessary sample dilution but the C–H vibration was consistently observed in three independent runs, but not for O–H or N–H. Therefore, the *in situ* DRIFTS experiments demonstrate the NAC-800 can activate the H₂ by forming the C–H bonds rather than O–H and N–H on its surface, as verified by DFT calculations. **All spectra in the *in situ* DRIFTS study are described and added in the revised manuscript.**

Supplementary Fig. 6 The *in situ* DRIFTS spectra of NAC-800 (diluted 20× with KBr). (a) and (b) are collected under He at 30 and 240 °C, respectively, using the spectra of KBr under He at the corresponding temperature as the background. (c) and (d) are the time-resolved spectra at 240 °C under flowing He and H₂, respectively, using the spectrum of NAC-800 (diluted 20× with KBr) under He at 240 °C as the background. For (c) and (d), time was recorded either after reaching the desired temperature (for He) or after switching gas (for H₂). The DRIFTS spectrum of NAC-800 under the flow of He at 30 °C is primarily composed of graphitic *sp*² domains (1578-1593 cm⁻¹) and C-N bond (1269-1290 cm⁻¹)⁴. No change to the spectra was observed after increasing the temperature to 240 °C. *In situ* DRIFTS study shows the formation of C-H bond (at ~2930 cm⁻¹)⁵ rather than O-H or N-H at 240 °C under the flow of H₂.⁴

We also attempted ¹³C solid-state NMR study of NAC-800. To spin this sample under MAS, we dilute the sample with sulfur in a 1:4 ratio due to its highly conductive nature (Grobelyny, J. et al. *Synth. Met* **29**, 97-102 (1989); Wang, Z. et al. *Carbon* **131**, 102-110 (2018)). We collected the ¹³C{¹H} CPMAS spectrum of NAC-800 (Fig. R1) with 24 h of signal accumulation due to low hydrogen content. Only one broad signal was observed centered around 120 ppm assigned as the aromatic carbons (Snape, C.E. et al. *Fuel* **68**, 547-560 (1989); Wang, Z. et al. *Carbon* **131**, 102-110 (2018)), suggesting that most carbons are in the graphitic N-heterocycle structures, in agreement with the DRIFTS study. Since no specific information on the active sites can be learned, the spectrum is not added in the revised manuscript.

Fig. R1 $^{13}\text{C}\{^1\text{H}\}$ CPMAS spectrum of NAC-800 at room temperature.

Overall, in view of the above criticisms I do not recommend this paper for publication in the current form. To be suitable for Nature Communications, major revisions are necessary.

Response: The manuscript was revised thoroughly based on the comments and suggestions. During the revision, we have performed several control experiments to exclude the disturbance of metal impurities for the metal-free hydrogenation (Supplementary Table 6), used isotopic exchange experiment to rule out the possibility of 2-PrOH as a hydrogen source (Supplementary Figs. 29 and 30), re-deconvoluted all the N1s spectra in NACs to verify the accuracy of the second-order polynomial relationship (Supplementary Tables 2 and 3; Fig. 1f; Supplementary Figs. 9 and 22), and performed *in situ* DRIFTS study (Supplementary Fig. 6). In addition, the possible mechanisms (Fig. 5d; Supplementary Fig. 19) were verified by DFT calculations; more details of the DFT calculations were included as well (Supplementary Figs. 24-26 and Supplementary Table 12). We appreciate your comments and suggestions, which are now addressed in this revised manuscript.

Reviewer #3

The present study demonstrates that graphitic mesoporous carbon-nitride materials synthesized following literature recipe can efficiently catalyze hydrogenation, dehydrogenation and hydrodeoxygenation of selected compounds. The synthesis of the catalysts were carefully controlled and the intermediate states and compositions were accurately characterized. N-sites with different oxidation states were identified by XPS (pyridinic, pyrrolic, graphitic and N-O). It has been shown that the catalysts can sufficiently chemisorb H₂. The selected catalyst (NAC-800) could hydrodeoxygenate the model PPE molecule and a considerably high selectivity toward the C-O attack instead of the aryl rings has been shown. This is a clear advantage over many typical TM catalysts. The scope of the hydrodeoxygenation includes C-O, carbonyl, olefinic, alkynyl and NO₂ groups. The same catalyst showed remarkable activity in dehydrogenation and hydrogenation as well. Thorough kinetic analysis has been performed and the kinetic order of the reactant PPE, H₂ and catalyst has been determined. The catalytic mechanism was addressed by the kinetics measurements complemented by periodic DFT calculations. The calculations revealed catalytically feasible route (low TS energetically neutral pathway) and excluded kinetically (high TS) or thermodynamically unfavorable (too stable chemisorption) routes. A remarkable accord in experiment and theory is the plausible explanation of the role of the graphitic N-pairs in the H₂ activation. In my opinion the study provides very new ideas for the metal-free H₂ activation field and also they have disclosed this new type of catalysis for the family of mesoporous carbon-nitrid materials. The study has addressed important catalysis issues (thorough characterizations, different kinetic measurements) and provided convincing mechanistic insights for the operation of the catalysts. I propose the publication of the study. However, I have the following issues which need to be addressed:

(1) Regarding selectivities: it is shown that NAC-800 boosts orthogonal processes: hydrogenation and dehydrogenation. Hence it would be important/useful to define circumstances or conditions or parameters how to obtain one or the other process during operation.

Response:

Thank you for your comments. The hydrogenation and dehydrogenation (such as the interconversion of quinoline and tetrahydroquinoline (THQ)) can be at equilibrium at a given temperature. From a thermodynamic point of view, the equilibrium of the reaction will shift to dehydrogenation at high temperatures because dehydrogenation is of positive enthalpy and entropy. In addition, reaction pressure could shift the hydrogenation/dehydrogenation equilibrium as well.

This temperature dependence was confirmed by the hydrogenation of quinoline and dehydrogenation of THQ in batch reactors for 4 h (Supplementary Table 10). While the quinoline hydrogenation at 140 °C with 10 bar H₂ gives a 5% conversion, increasing the temperature to 230 °C causes a conversion drop to <1% even with the presence of 20 bar H₂. Therefore, the hydrogenation reaction can be shut off at high temperatures. For the dehydrogenation of THQ, the catalyst achieves a 6% conversion to quinoline at 140 °C. At 230 °C, the conversion increases to 10%, even in the presence of 20 bar H₂. The temperature-

dependent interconversion of quinoline and tetrahydroquinoline agree with the thermodynamic prediction.

Supplementary Table 10 Metal-free dehydrogenation of THQ or hydrogenation of quinoline over NAC-800 catalyst.

entry	reactant	equilibrium shift to	temperature (°C)	gas	pressure (bar)	conversion (%)
1	THQ	Dehydrogenation	140	H ₂	10	6
2	quinoline	Hydrogenation	140	H ₂	10	5
3	THQ	Dehydrogenation	230	H ₂	20	10
4	quinoline	Hydrogenation	230	H ₂	20	<1

Results for all reactions are now added into the supplementary information as Supplementary Table 10. The following sentences are now added in the manuscript:

“The dehydrogenation reactions of THQ are much slower under high pressure of H₂ (Supplementary Table 10). Under 10 bar H₂, a 6% conversion to quinoline was observed at 140 °C in 2 h, while the conversion increased slightly to 10% at 230 °C. The reverse reaction, quinoline hydrogenation, can also be catalyzed by NAC-800, showing a 5% conversion at 140 °C with 10 bar H₂. However, increasing the temperature to 230 °C caused the conversion to drop to <1% even with 20 bar H₂. Thus, the hydrogenation reaction of quinoline to THQ over NAC-800 is greatly temperature-dependent and can be shut off at higher temperatures.”

(2) regarding the calculations: the NEB profiles should be shown in the SI or at least some information about the imaginary frequencies describing the activation processes probed by the computations should be given.

Response:

In the supplementary material, we now add the following figure to show the NEB profiles (Supplementary Fig. 26) as well as a table including the imaginary frequencies of the transition states (Supplementary Table 9). Note the NEB structures were adopted to get the transition states using the dimer method, so the values of the activation barrier from the NEB are very close to but not the same as the values reported in the main text, where the transition states were further optimized using the dimer method and verified by calculating the vibrational frequencies.

Supplementary Fig. 26 Reaction profiles from DFT NEB calculations. Note these calculations were later used for further optimization to get the transition states using the Dimer method, so the values of the activation barrier from the NEB are very close to but not the same as the values reported in the main text, where the transition states were further optimized using the Dimer method and verified by calculating the vibrational frequencies.

Supplementary Table 12 DFT-calculated imaginary frequencies of the transition states.

Configurations	Imaginary frequencies (THz)
a12	31.3
a13	54.9
b12	47.5
c12	38.5
d12	48.0
d13	25.3
e12	43.3
e13	24.8

REVIEWERS' COMMENTS:

Reviewer #1 (Remarks to the Author):

Ref. The manuscript "Transition Metal-Like Carbocatalyst" by Wang et al, NCOMMS-20-07371A
The authors prepared a careful response to the comments raised by the referees, including myself.
However, the activity of the catalyst in the targeted process (reactions carried out between 32 and 80h)
with modest selectivities in the most important reaction products is not competitive. Therefore I am still
not convinced that this manuscript is at the level of exigency reached by Nature Communications.

Reviewer #2 (Remarks to the Author):

The manuscript has been improved and I support its publication in Nature Communications.

Reviewer #3 (Remarks to the Author):

In my opinion the authors have addressed satisfactorily all the issues that were raised in my evaluation
and also by the other two reviewers. I find their full response very detailed, convincing and also from
other perspective intellectually very much entertaining. I propose the publication of the study.

Reviewer #1 (Remarks to the Author):

The authors prepared a careful response to the comments raised by the referees, including myself. However, the activity of the catalyst in the targeted process (reactions carried out between 32 and 80h) with modest selectivities in the most important reaction products is not competitive. Therefore I am still not convinced that this manuscript is at the level of exigency reached by Nature Communications.

Response:

We recognize that NACs are not as active as transition metals when comparing the rates. However, distinct from metal catalysts, NACs can prevent over-hydrogenation of phenyl ring, which is unique and highly desirable in view of both efficient H₂ utilization and atom economy. The selectivity was defined and described in our previous response.

Our work, combining both experimental and theoretical endeavors, demonstrate that nitrogen assemblies rather than isolated N atoms are the active sites of NAC catalysts, enabling direct C-O hydrogenolysis, which has never been reported for metal-free catalysts to our best knowledge. In addition, NACs can also efficiently catalyze the non-oxidative dehydrogenation of ethylbenzene and tetrahydroquinoline and the selective hydrogenation of unsaturated functional groups in aromatics. Besides, nitrogen assemblies can be a new member of surface active sites for the carbon-based materials, in addition to existing ones, including carbon defects [Wen, G. et al. *Angew. Chem. Int. Ed.*, **54**, 4105–4109 (2015)], frustrated Lewis pairs [Primo, A. et al. *Nat. Commun.* **5**, 5291 (2014)], carbonyls [Yang, H. et al. *Nat. Commun.* **6**, 6478 (2015)], isolated N sites [Kong, X. et al. *Energy Environ. Sci.* **6**, 3260-3266 (2013)], and 5-membered rings [Jia, Y. et al. *Nat. Catal.* **2**, 688–695 (2019)]. Therefore, we believe this work will provide new and valuable guidance for the design and synthesis of metal-free catalysts with unique catalytic performance.

Reviewer #2 (Remarks to the Author):

The manuscript has been improved and I support its publication in Nature Communications.

Response: We thank the reviewer for the comment.

Reviewer #3 (Remarks to the Author):

In my opinion the authors have addressed satisfactorily all the issues that were raised in my evaluation and also by the other two reviewers. I find their full response very detailed, convincing and also from other perspective intellectually very much entertaining. I propose the publication of the study.

Response: We thank the reviewer for the comment.